# Ask more, know better: Reinforce-Learned Prompt Questions for Decision Making with Large Language Models

## Abstract

Large language models (LLMs) demonstrate their promise in tackling complicated practical challenges by combining action-based policies with chain of thought (CoT) reasoning. Having high-quality prompts on hand, however, is vital to the framework's effectiveness. Currently, these prompts are handcrafted utilising extensive human labor, resulting in CoT policies that frequently fail to generalise. Human intervention is also required in order to develop grounding functions that ensure low-level controllers appropriately process CoT reasoning. In this paper, we take the first step towards a fully integrated end-to-end framework for task-solving in real settings employing complicated reasoning. To that purpose, we offer a new leader-follower bilevel framework capable of learning to ask relevant questions (prompts) and subsequently undertaking reasoning to guide the learning of actions to be performed in an environment. A good prompt should make introspective revisions based on historical findings, leading the CoT to consider the anticipated goals. A prompt-generator policy has its own aim in our system, allowing it to adapt to the action policy and automatically root the CoT process towards outputs that lead to decisive, high-performing actions. Meanwhile, the action policy is learning how to use the CoT outputs to take specific actions. Our empirical data reveal that our system outperforms leading methods in agent learning benchmarks such as Overcooked and FourRoom.

## 1 Introduction

Large language models (LLMs) with Chain-of-thought (CoT) prompts (Wei et al., 2022; Wang et al., 2022) have achieved impressive performance improvements for solving complex natural language processing (NLP) tasks. Moreover, techniques such as reward incentives (Yao et al., 2022; Hao et al., 2023) have been shown to enhance the quality of Chain-of-Thought prompts for addressing intricate tasks. Two notable approaches, Tree-of-Thought (ToT) (Yao et al., 2023) and Reasoning via Planning (RAP) (Hao et al., 2023), have emerged to be useful techniques that leverage LLM-generated reward functions for guiding the step-by-step problem-solving process. With the increasing reasoning capabilities of CoT, the reasoning outputs of LLMs can be used to provide useful 'thought' inputs to policies that perform tasks in practical environments. This involvement of CoT reasoning has given rise to the promise of unlocking the power of LLMs to be able to assist in performing complex automated reasoning and acting in real-world environments.

While LLMs such as ChatGPT possess a wealth of human knowledge, in general, current methods (Yao et al., 2023; Hao et al., 2023) heavily depend on meticulously crafted prompts designed by humans for each specific task. Moreover, the performance of CoT reasoning can be sensitive to the quality of the prompt input — poor prompts provided even to powerful LLMs are unlikely to generate useful CoT outputs. Additionally, despite the obvious potential of using CoT reasoning for guiding a low-level control policy, human-intelligible CoT reasoning can often be ambiguous for a downstream control policy, such as a rule-based planning method (Zhang et al., 2023; Shah et al., 2023) and an action policy implemented by a reinforcement learning (RL) algorithm (Carta et al., 2023). As such, a natural consideration is for the need to generate CoT outputs that are interpretable to the action policy and, provably reduce the uncertainty of the action policy. Therefore, the ambition of embedding CoT

reasoning within a generalist artifical intelligence (AI) framework has produced a series of critical challenges that have yet to be fully resolved.

In this paper, we take the first step towards a fully unified LLM framework that learns to perform complex tasks. In order to achieve this goal, both the prompt design and the policy that outputs actions to be executed have to be sufficiently flexible and useful so as to adapt to the current task at hand. Tackling this challenge necessitates learning both to generate appropriate questions (a.k.a. prompts) given environment observations as well as learning how to perform actions that enable the task to be solved. To this end, we introduce a decision-making framework which learns to ask pertinent *questions* or perform introspection, performs CoT reasoning and then learns to take the best actions in the environment. The first component of the framework is enacted by a *prompt-generation policy* that learns a suitable prompt question given the current challenge and overall task and given its observations of the environment. These prompts serve as inputs to a CoT process; this then allows the framework to perform desired and complex reasoning given the prompt. The CoT *thoughts* are then inserted into the action-policy that learns to find solutions to tasks that may require both interaction experience and human knowledge embedded in CoT reasoning to solve.

Learning how to generate in-demand prompts for the CoT process produces formidable challenges. One such challenge is to ensure that the resulting CoT thoughts enhance the performance of an action policy. Departing from a fixed set of pre-selected, human-crafted prompts and learning to find useful prompts to be fed into the CoT process presents an important challenge. Specifically, ensuring that the resulting CoT thoughts improve the performance of an action-policy that can solve the task. We resolve this challenge by designing a *leader-follower Bilevel* structure, called Bilevel-LLM and illustrated in Figure 1, that generates mutually adaptive policies. Each policy is endowed with its own objective — the prompt-generation policy observes the effect of its prompt on the action policy and learns to generate useful prompts, and subsequent CoT outputs that are correctly interpreted. In particular, the prompts and CoT output are chosen so as to minimise the uncertainty of the action policy i.e. the prompt-generation policy chooses prompts that minimise the entropy of the action-policy. The action policy, on the other hand, learns to maximise the environmental reward while taking into account the outputs of the CoT process. Ultimately, the generated thoughts serve to learn a more effective action policy, providing additional information beyond the observation of the environment. These natural language insights embody human knowledge, reducing the need for redundant exploration compared to traditional RL algorithms, which typically require extensive exploration of specific environments to train a competent agent.

In numerous task environments, expert prompt data for the task is available, such as a well-defined set of subtasks (Shah et al., 2023). Making use of this in *decision-making* tasks requires prompts that induce CoT reasoning for performing desirable actions at *each state*. Nevertheless, often, the information in expert prompt sets is not refined to capture useful specifics at the state level producing a challenge of how to select the appropriate prompt at a given state. In environments where such prompt candidates are not available, the challenge becomes autonomously generating useful prompts using only the environment observations. In Sec. 4, we demonstrate Bilevel-LLM is capable of tackling each of these challenges. First, we demonstrate that Bilevel-LLM successfully learns to select, from a global set of candidate prompts, the best prompt for each state. We then demonstrate that in problem settings where prompt candidates are not available, Bilevel-LLM successfully generates desirable prompts at each state entirely from state observations.

The contributions of this paper can be summarised as follows:
• A new framework for auto-generation of prompts for decision-making tasks. An integral component is a prompt-policy or *prompt-generator* which is trained by our framework to generate prompts that induce low uncertainty in the action-policy which receives thoughts generated by CoT reasoning triggered by the prompts from the prompt-generator. Therefore, the prompt-generator (and hence CoT process) behaves adaptively toward the needs of the action-policy.
• A chain-of-thought generation framework in which the thought output of the CoT process are used to guide a policy that takes actions within an environment in order to solve practical tasks. This leverages the benefits of natural language models and CoT reasoning that encapsulate worldly experience and the capacity for deductive reasoning while efficiently tuning the thought pipeline process by tuning the prompt generation policy.
• Prompt-tuning plus learning of LLM input-based policy that acts in environment (dual framework).
• A new bilevel learning algorithm that uses natural language to guide what actions and finds prompts for this desired textual guidance.

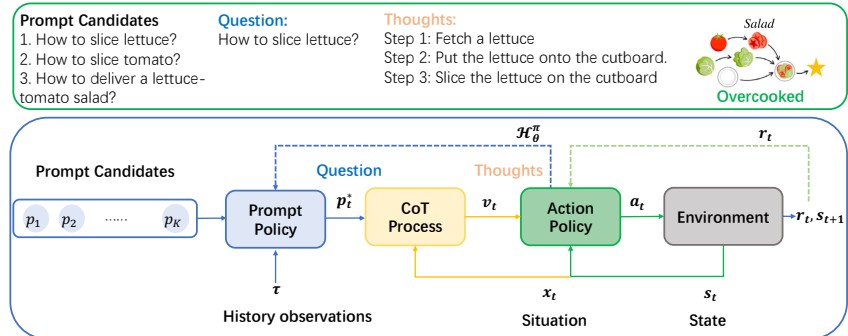

Figure 1: *Top:* Example of the workflow from Prompt candidates to CoT reasoning. *Bottom:* The illustration of our bilevel optimisation framework.

## 2 PROBLEM FORMULATION

In this setting, an agent aims to solve some task by performing a sequence of actions in an environment. Formally, the problem is described by a partially observable Markov decision process (POMDP), which is defined by the following tuple $\langle \mathcal{S}, \mathcal{A}, P, \mathcal{O}, T, \mathcal{R}, \gamma \rangle$ where $\mathcal{S}$ is the finite set of environment states, $\mathcal{A}$ is the set of actions for the agent, $P : \mathcal{S} \times \mathcal{A} \to \Delta(\mathcal{S})$ is the state transition kernel for the environment, $\mathcal{O}$ is the finite set of observations. The states and observations can be represented as symbolic vectors, which can be translated into text descriptions conveying the information in the vectors. The function $\mathcal{R} : \mathcal{S} \times \mathcal{A} \to \Delta(\mathbb{R})$ is the reward function, which returns a scalar reward conditioned on a state-action pair whose realisation at time step $t$ we denote by $r_t \sim R$ and lastly, $\gamma \in [0, 1]$ is the discount factor. We introduce an additional variable $x_t \in X$ contained in the *situation space* $X$. The variable $x_t$ represents observed primary information that can be encoded as text that can help a CoT reasoning LLM to generate task relevant thoughts. Lastly, the observation function is $T : \mathcal{S} \times \mathcal{A} \times X \to \mathcal{O}$ which is a mapping from the environment state, action and situation to the observation set of the agent. In challenging problems, standard methods such as RL struggle to solve these tasks in a sample efficient way. In order to solve complex decision-problems, an agent may be challenged with needing to perform deductive reasoning in order to resolve the challenge of finding an optimal policy.

To tackle these challenges, we equip the agent with both a dual LLM structure that enables the agent to first, generate its own pertinent prompts from its observations of the current. Then, using these prompts, perform CoT reasoning to perform complex reasoning about the best course of action. Lastly, an action is taken in the environment. The framework can therefore be split into three components: a *prompt-generating* policy $\pi_\phi : (\mathcal{O})^{j<\infty} \to \mathcal{T}$. This policy learns to generate prompts after observing (a window of) $j < \infty$ observations and outputs a thought in textual thought space[1] Second, a *thought reasoning* policy $\pi^{\mathrm{re}} : \mathcal{S} \to \mathcal{T}$ — an LLM that reasons about the task at that particular state by performing CoT to generate a thought output. Denote $\mathcal{V}$ is the vocabulary (with finite words in it). Each thought $t \in \mathcal{T} \in \mathcal{V}^M$ is described as a sentence with $M < \infty$ tokens in it where $\mathcal{T}$ is the set of thoughts. The thought reasoning policy $\pi^{\mathrm{re}}$ does step-by-step thought reasoning, e.g. Opening the box requires finding the key and then unlocking it in natural language space. The CoT reasoning is performed by an LLM, since Bilevel-LLM is a plug & play framework, any choice of LLM can be used to perform the CoT reasoning (in our experiments we use GPT3.5 [2]). Lastly, an *action-policy* $\pi_\theta : \mathcal{O} \times \mathcal{T} \to \Delta(\mathcal{A})$. The action-policy makes an observation of the environment and takes the CoT thought as an input then executes actions in the environment.

Therefore, at times $t = 0, 1, \ldots$, a prompt $p_t$ is generated by the prompt generation policy i.e. $p_t \sim \pi_\phi(\cdot | o_t, \ldots, o_{t-j \wedge 0})$. The prompt is then used by an LLM to trigger a CoT process whose output is a thought $v_t \in \mathcal{T}$. Last, the action-agent samples an action from its policy $a_t \sim \pi_\theta(\cdot | o_t, v_t)$, where $t$ is the time immediately after querying the thought reasoning policy $\pi^{re}$ at time step $t$. Therefore, sequence of events proceeds as follows:
**1.** At time $t = 0, 1, \ldots$ the system is at an environment state $s_t \in \mathcal{S}$.
**2.** A prompt $p_t$ is produced by the prompt generation policy i.e. $p_t \sim \pi_\phi(\cdot | o_t, \ldots, o_{t-j \wedge 0})$.

---

[1]The Markov property is ensured by setting $j = 0$.
[2]The version of GPT3.5 in this work is GPT3.5-turbo

**3.** An action $a_t \sim \pi_\theta(\cdot|o_t, v_t)$ is taken given the output of the CoT process $v_t \sim \pi^{\text{re}}(p_t, x_t)$, $x_t$ is the observed situation of the current state.
**4.** The environment state transitions according to $s_{t+1} \sim P(\cdot|s_t, a_t)$. Figure 6 shows a step by step inference example of Bilevel-LLM on the Overcooked task.

To tackle the problem of learning how to generate prompts while learning the action-policy, we structure the problem as a leader-follower *bilevel optimisation* Colson et al. (2007). This allows the prompt-generator policy to learn how its actions affect the action-policy while action-policy and prompt-generator policy learn concurrently. In this way, the prompt-generator policy alters its output to produce desirable actions from the action-policy while the action-policy learns both how to interpret the CoT outputs and take desirable actions. Since LLMs already contain a vast amount of world knowledge, we here fix the LLM that performs the CoT reasoning, that is we assume that $\pi^{\text{re}}$ is pretrained and fixed. We update the prompt-generator policy and action-policy. The aim of the prompt-generator policy is to generate prompts minimise the uncertainty of the action policy action. The optimisation objective can be expressed as a bilevel optimisation problem:

$$(\pi_\theta^*, \pi_\phi^*) \in \underset{(\pi_\theta, \pi_\phi) \in \Pi_\theta \times \Pi_\phi}{\arg\max} \mathbb{E}_{\pi_\theta, \pi_\phi, \pi^{\text{re}}} \left[ -\sum_{t \geq 0} \gamma^t \mathcal{H}^{\pi_\theta}(y_t) | y_t = (o_t, v_t), v_t \sim \pi^{\text{re}}(p_t) \right]$$

$$\text{s.t. } \pi_\theta^* \in \underset{\pi_\theta \in \Pi_\theta}{\arg\max} \mathbb{E}_{\pi_\theta, \pi_\phi, \pi^{\text{re}}} \left[ \sum_{t \geq 0} \gamma_I^t r_t | p_t \sim \pi_\phi \right], \ \forall \pi_\phi \in \Pi_\phi, \forall \pi^{\text{re}} \in \Pi^{\text{re}}, \tag{1}$$

where $\mathcal{H}^{\pi_\theta}(y_t) = \sum_{a_t \in \mathcal{A}} \pi_\theta(a_t|y_t) \log \pi_\theta(a_t|y_t)$, $y_t = (o_t, v_t)$ which is the entropy of the policy $\pi_\theta$ and $\gamma_I, \gamma \in [0, 1)$ are the discount factors and $r_t \sim \mathcal{R}$. Note that the bilevel aspect incorporates the nested nature of the optimisation Colson et al. (2007); Dempe (2002) — in order to find the optimal prompt, the prompt-generator policy must take into account the anticipated behaviour of both the LLM $\pi^{\text{re}}$ and the action policy $\pi_\theta$ and thereafter make its choice accordingly.

## 3 METHODOLOGY

In this section we describe the training procedure of proposed the Bilevel framework. The prompt generation policy is optimised via the policy gradient with the behavior of action policy as a reward. The action policy is served by an LLM with PPO updater, which benefits from avoiding human-crafted engineering by grounding the CoT reasoning to executable actions. In the bilevel framework, the prompt generation policy and action policy are concurrently optimised until convergence.

CoT reasoning with LLMs has proven to be effective in aiding decision-making when well-designed prompts are used (Zhang et al., 2023; Park et al., 2023). However, the quality of CoT reasoning heavily depends on the quality of prompts, which are typically manually designed by humans (Zhang et al., 2023; Shah et al., 2023). In traditional Natural Language Processing (NLP) tasks such as sentiment classification (Pang et al., 2002) and news classification (Rana et al., 2014), prompts are usually provided through sets of input-output pairs. Unlike these NLP tasks with clearly defined input-output examples, the desired format of prompts varies across different decision-making tasks, often requiring substantial manual engineering.

**Prompt generation policy training via policy gradient.** Due to the difficulty of training a model that is able to generate reasonable prompts from scratch automatically, we alternatively use pre-defined prompt candidates which can be obtained by human deliberately writing or being generated by GPT3.5. We have conducted an experiment about using GPT3.5 to generate prompts, where the task description, state, and abstracted state situation are inputted into GPT3.5 to produce some simple prompt questions about how to achieve the goal. Examples of prompt candidates generated from GPT3.5 are shown in Appendix 9.2.

While human assistance is engaged in generating prompt candidates, our work focuses solely on generating prompts about the critical subtasks, similar to the approach in (Shah et al., 2023), but less extensive than in (Zhang et al., 2023; Park et al., 2023), where human-designed prompt formats are required for the entire decision-making process, encompassing ensuring a subgoal, thinking and acting.

---

**Algorithm 1** Bilevel-LLM

---

**Input:** Initialise parameters of policies $\pi_\theta$, $\pi_\phi$. Set the data buffer $D = \emptyset$.
**Output:** $\pi_\theta^*$, and $\pi_\phi^*$.

1: **while** not done **do**
2:     #Rollout trajectories with $\pi_\theta, \pi^{re}, \pi_\phi$.
3:     **for** $i = 1, 2, .., \text{step}$ **do**
4:         Generate prompt given historical observation : $p_t \sim \pi_\phi(\cdot|o_t, \ldots, o_{t-j\wedge 0})$
5:         Perform CoT process given prompt $p_t$, generate terminal CoT thought $v_t$
6:         # Note that $t$ is the time after querying the thought reasoning policy $\pi^{re}$.
7:         Sample action according to the thought and observation: $a_t \sim \pi_\theta(\cdot|o_t, v_t)$
8:         Apply action $a_t$ to the environment, sample reward $r_t \sim \mathcal{R}(s_t, a_t)$. The next state according to $s_{t+1} \sim \mathcal{P}(\cdot|s_t, a_t)$. The next step observation is $o_{t+1} \sim T(\cdots|s_{t+1})$.
9:         Calculate the entropy of the action policy $h_t = \mathcal{H}(\pi_\theta(\cdot|s_t, v_t))$
10:       Add to data buffer: $D = D \cup (o_t, p_t, v_t, a_t, r_t, h_t, o_{t+1})$
11:     **end for**
12:     **for** Epochs and Batch numbers **do**
13:         Sample a batch of data $d$ from $D$.
14:         Update the prompt generation policy $\pi_\phi$ by policy gradient following Eq. (**??**),(2).
15:     **end for**
16:     **for** Epochs and Batch numbers **do**
17:         Sample a batch of data $d$ from $D$.
18:         Update the action policy $\pi_\theta$ via PPO to optimise Eq. (3).
19:     **end for**
20: **end while**

---

With a prompt candidate set $\mathcal{P} = \{p_1, p_2, \cdots p_K\}$, we train a prompt generation policy $\pi_\phi(\cdot|o_t, \ldots, o_{t-j\wedge 0})$ over the prompt candidates according to historical observation $o_t, \ldots, o_{t-j\wedge 0}$. Each of these natural language prompt candidates can be represented as a high-dimensional vector using a pre-trained and frozen Bert Devlin et al. (2018) model. Denote the embedding of prompt candidate $p_i$ as $e_i$ and the embedding of the historical observations $(o_t, \ldots, o_{t-j\wedge 0})$ as $e_o = \mathcal{E}(o_t, \ldots, o_{t-j\wedge 0})$ with the encoder $\mathcal{E}(\cdot)$. The prompts' embedding and the observations' embedding are projected into the same vector space. Denote the mapped embedding as $\hat{e}_i = \mathcal{M}_p(e_i), \forall i = 1 \cdots K$, $\hat{e}_o = \mathcal{M}_o(e_o)$, where $\mathcal{M}_p$ and $\mathcal{M}_o$ are projectors for prompts and observation sequence respectively. During the decision-making process, the prompt policy estimates the probability of selecting a prompt candidate $p_i$ based on the similarity between the prompt candidate embedding $\hat{e}_i$ and the historical observation sequence's embedding $\hat{e}_o$. The prompt policy is updated via the policy gradient with the minus entropy of action policy as a reward incentive and parameters of the observation encoder $\mathcal{E}$ and projectors $\mathcal{M}_p$, $\mathcal{M}_o$ are trainable. The detailed procedure is described as below:

• For a given decision-making task, we employ GPT-3.5, along with the provided task description, to generate appropriate prompt candidate sets. As a second case, we used human-crafted assists to generate valuable prompt candidates.
• With these $K$ prompts, the prompt generation policy is updated with the objective of maximising the minus action-policy entropy. The objective function is given by:

$$J_\phi(y|\pi_\theta, \pi_\phi, \pi^{\text{re}}) = \mathbb{E}_{\pi_\theta, \pi_\phi, \pi^{\text{re}}}\left[-\sum_{t \geq 0} \gamma^t \mathcal{H}^{\pi_\theta}(y_t)|y_t = (o_t, v_t), v_t \sim \pi^{\text{re}}(p_t, x_t), y_0 = y\right]$$

• We use a policy gradient (Lu et al., 2022) to optimise the prompt generation policy which obeys the following expression:

$$\nabla_\phi J(y|\pi_\theta, \pi_\phi, \pi^{re}) \approx \frac{1}{N}\sum_{t \geq 0} \nabla_\phi \log \pi_\phi(p_t|o_t, \ldots, o_{t-j\wedge 0})\hat{R}_t^o(\tau). \tag{2}$$

The prompt generation policy $\pi_\phi$ is updated according to $N$ sampled trajectories from polices $\pi_\theta$, $\pi_\phi$, and $\pi^{re}$. We denote $\hat{R}_t^o(\tau) = -\sum_{i \geq t}\left[\gamma^{i-t}\mathcal{H}^{\pi_\theta}(y_i)|y_{i+} = (o_i, v_i), v_i \sim \pi^{\text{re}}(p_i, x_i)\right]$ as the return-to-go from step $t$ to the end for the outer loop.

**CoT reasoning with Prompts.** With the selected prompt $p_t$ sampled from the prompt candidate set, the CoT reasoning information is obtained by $\upsilon_t \sim \pi^{re}(\cdot|p_t, x_t)$, where the CoT reasoning policy $\pi^{re}$ is served by an LLM such as GPT3.5. The motivation of integrating the CoT reasoning into our bilevel framework, is we hope to use the prior human experts' knowledge to provide a high-level guideline of how to solve complicated decision-making tasks. For example, as shown in Figure 1, in Overcooked game, the CoT LLM can generate a sequence of intermediate steps need to be done with a prompt about the subtasks "how to slice lettuce" given. About how to finish the intermediate steps, previous studies (Zhang et al., 2023; Shah et al., 2023) rely on some hand-crafted and rule-based strategies to understand CoT reasoning and perform actions. In this work, we fed CoT reasoning into the action policy served by a small LM to automatically interpret CoT outputs. To reduce the time and cost associated with frequent queries to GPT-3.5, we abstract situations to represent states and stored CoT reasoning outputs for the same situations. For example, in the case of two distinct states, even though the agent may be in different positions and neither state involves holding lettuce, they are considered part of the same situation because the steps to slice lettuce remain the same: picking up a lettuce, placing it on the cutting board, and then proceeding to slice it.

**Action policy training via PPO with LLM.** Existing works (Jang et al., 2021; Carta et al., 2023) utilise LLMs as the action policy and fine-tune these LLMs to adapt to decision-making tasks, taking advantage of the comprehensive capabilities of LLMs. In our work, we also utilise an LLM as the action policy. Within our framework, in addition to considering the textual observations provided by the environment, we also incorporate additional CoT reasoning from GPT-3.5 when performing actions. To ground the outputs from the action LLM into executable actions, we fine-tune the action LLM, denoted as $\pi_\theta$, using Proximal Policy optimisation (PPO) (Schulman et al., 2017). The objective of the action policy is to maximise environment return:

$$\arg\max_\theta \mathbb{E}_{\pi_\theta, \pi_\phi, \pi^{re}} \left[ \sum_{t \geq 0} \gamma_I^t r_t | a_t \sim \pi_\theta, \upsilon_t \sim \pi^{re}, p_t \sim \pi_\phi \right]. \tag{3}$$

During the training phase of the action policy $\pi_\theta$, we freeze the prompt generation and CoT reasoning policies and finetune the action policy with collected trajectories. Additionally, we use the pre-trained LLM, Flan-T5 small (Rae et al., 2021) with parameters less than one billion as the action policy.

**Bilevel Optimisation.** In our leader-follower Bilevel LLM framework, the prompt generation policy and the action policies are trained alternately, with the other policy being kept frozen. On one hand, the prompt generation policy selects a prompt for the CoT reasoning LLM, which outputs are expected to be interpreted by the action policy. Thus, the goal of the prompt generation policy is to reduce the uncertainty of the action policy when it encounters challenging decisions. In practical terms, the objective is to minimise the entropy of the action policy. On the other hand, the action policy is trained to effectively solve specific decision-making tasks while benefiting from CoT reasoning and the experience gathered during exploration. The overall training process of the Bilevel framework is detailed in Algorithm 1.

## 4 EXPERIMENTS

In this section, we verify the effectiveness of Bilevel framework on three environments. Further details on experimental settings and ablation studies can be found in the Appendix. We perform our empirical examinations on the following three environments:

**ChainWorld.** The ChainWorld game contains a linear sequence of states and the available actions for the agent are go left or go right. The agent gains a reward $100$ at a random end of the chain and $-5$ at the other end, with $-1$ penalty for each move. At each episode, the award randomly appears on the left or right end, and the initial position of the agent is randomized, except for the ends. There are two situations corresponding to different sides with high rewards. We consider two settings: *ChainWorld(Full)*, where full observation of the situation and position information are provided, and *ChainWorld(Partial)*, where only partial observation of the agent's position is available. In the case of ChainWorld(Partial), since the position with a reward of $100$ is randomized, the agent must learn to make decisions based on historical trajectory information.

**FourRoom.** In this game, four rooms are circularly interconnected by four hallways, and an agent needs to reach a goal in these rooms. The agent's position and the goal position are randomly initialized within these four rooms at the start of each game. The objective for the agent is to reach

Figure 2: Results of comparison with baselines. We plot the mean and standard error of nomalized reward over 5 seeds for trainable baselines, and over 20 episodes for GPT3.5 baselines. We normalize the cumulative rewards within the range [0,1] and calculate the Area Under the Curve (AUC) by averaging over all episodes during training or inference.

the goal as fast as possible. During each step, the agent receives two types of information: a global observation of the goal's position and its own current position, and a partial observation of the hallways within its current room. Based on this information, the agent decides and moves one cell.
**Overcooked.** Overcooked has a discrete action space with 4 directions; North, East, South, and West. This game has the following items: tomato, lettuce, two cut boards, and two plates. An agent can pick up food items and chop them on the cutboard, or place the chopped food on a plate. The goal is to make and deliver the desired meal. We have designed candidate prompts with which we also get CoT examples from GPT3.5. We consider two different recipes in Overcooked: delivering a chopped tomato and delivering a tomato-lettuce salad. We also consider a large layout with a map size of $7 \times 7$ and a recipe of tomato-lettuce salad. For the common layouts, the map size is $5 \times 4$.

We compare Bilevel-LLM with two trainable baselines and two baselines that directly prompt GPT3.5 to perform actions, namely: **GFlan** (Carta et al., 2023). GFlan adopts the LLM Flan-T5 large as the foundation of action policy and optimises it via PPO algorithm. GFlan solely relies on textual observations as input and employs this information to estimate the conditional probabilities of the action tokens.
**Vanilla PPO** (Schulman et al., 2017). Unlike GFlan which leverages LLMs, Vanilla PPO employs conventional neutral networks such as MLPs as the backbone architecture and trains the action policy from scratch. We use the symbolic embedding of states as the input of action policy.
**GPT-3.5.** Previous studies (Yao et al., 2023; Hao et al., 2023) show that LLMs have impressive reasoning capability on natural language, we test the zero-shot decision-making capability of GPT-3.5 with task descriptions, textual context, and executable action candidates as input prompt and let GPT-3.5 infer the action at the current state.
**GPT3.5 with CoT prompt.** CoT prompts have the potential to substantially enhance the performance of CPT3.5 on complex reasoning tasks. Besides the inputs used in the GPT-3.5 setting, we further incorporate examples of human interactions with the environment or human-established task decompositions as a part of the input prompt and instruct GPT-3.5 to think step by step.
**Bilevel-LLM.** We propose the Bilevel LLM framework that integrates prompt generation, CoT reasoning, and action policies. Compared with GFlan, Bilevel-LLM leverages the additional prompt generation policy to select a suitable question for CoT reasoning LLM relying on historical observations. With the selected question, the CoT LLM can reason the human-like high-level solution of the question from human experts' knowledge contained in LLMs. The CoT reasoning, i.e, high-level solution can assist the action policy to solve the more complicated tasks.

**Comparison with baselines.** The results of comparisons with baselines are shown in Figure 2. Bilevel-LLM outperforms other baselines in all environments also exhibits a smaller standard error than the suboptimal GFlan. In most environments, GFlan also outperforms Vanilla PPO. This suggests that using a pre-trained LLM as the backbone of the action policy improves performance and training efficiency, due to the rich prior knowledge contained in the pre-trained LLM. In addition, for most environments, except for ChainWorld (Full), GPT-3.5 and GPT-3.5 (CoT) struggle to solve decision-making tasks effectively. This indicates that although GPT-3.5 is powerful in generating useful high-level task solutions (thoughts), it faces challenges in long-term decision-making processes due to the complexity of the world model and rules in the environment. Additionally, grounding the output of GPT-3.5 into executable actions proves to be challenging.

**Does Bilevel-LLM learn to automatically generate prompts?** We tested the case for when prompt information is not available which requires our method to learn its own prompts. *Bilevel-LLM-Auto* is displays the performance of Bilevel-LLM when the prompt candidates are automatically generated by GPT3.5 using only the observation and situation (which may be limited for a task) descriptions. As shown in Figure 3(c), Bilevel-LLM (combined with GPT3.5) automatically generated prompts that learn to successfully induce desirable prompts, CoT reasoning, and actions that solve the task well in the ChainWorld(Full). We also verify the automatically generated prompt candidates on Overcooked. As shown in Figure 4, *Bilevel-LLM-Auto* achieve similar rewards compared to

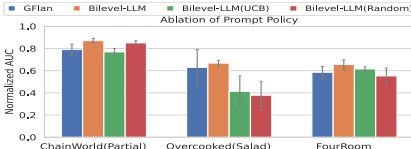
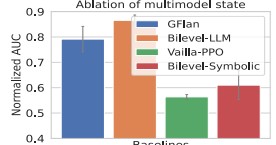
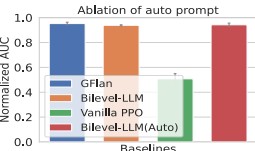

(a) Ablation on prompt policy    (b) Ablation on multimodal (c) Ablation on auto prompt

Figure 3: Ablation studies. (a) The effect of different prompt generation strategies. (b) Verficiation of the effectiveness of Bilevel-LLM under multimodal state representations on ChainWorld (Partial). (c)We verify the prompt candidates that are automatically generated from GPT3.5 with the state observations and situations given on ChainWorld(Full).

Bilevel-LLM with human-designed prompt candidates. In addition, Bilevel-LLM and *Bilevel-LLM-Auto* both outperform GFlan and exhibit lower variance. Specifically, after training the same number of episodes, Bilevel-LLM reaches a normalized reward around 1.0 but GFlan only reaches around 0.9. This suggests that CoT thoughts induced by appropriately selected prompts are helpful in solving complex decision-making tasks and that our framework is plug-and-play and can learn to use the automatically generated valuable prompts from GPT3.5. Examples of automatically generated prompts relying on state and situation descriptions can be found in Section 9.2 of the Appendix.

## 4.1 ABLATION STUDIES

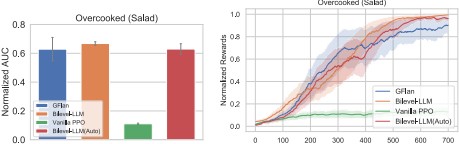

Figure 4: Automatically generate prompts on Overcooked(Salad). *Left:* Normalized AUC reward. *Right:* Rewards during training.

We conducted a series of ablation studies to confirm the usefulness of the components of Bilevel-LLM . In the following, we modified components of Bilevel-LLM in order to validate the following claims:

**Does the prompt policy with policy gradient improve performance?** In order to validate the claim that the prompts generated by Bilevel-LLM lead to improved performance, we tested Bilevel-LLM against the baseline *Bilevel-LLM (Random)*, which is Bilevel-LLM but with the prompt policy replaced so that we randomly select a prompt from the candidate set at each time step. In addition, *Bilevel-LLM (UCB)* views the prompt selection from a candidate set as the multi-armed bandit problem and uses Upper Confidence Bound (UCB) to select the prompt. In this setting, the UCB algorithm does not consider the historical observation but only relies on the environment rewards, i.e., the minus entropy of the action policy to select a prompt. In addition, the UCB counts are reset for each episode. As shown in Figure 3(a), Bilevel-LLM outperforms all other prompt policy versions on all environments. The bad performance of *Bilevel-LLM (UCB)* might be due to the lack of consideration of environmental state when performing prompt selection. **Does the entropy objective improve performance?** To validate the claim that the entropy objective leads to better performance we tested Bilevel-LLM against the baseline *Bilevel-LLM (Env)*, which replaces the negative entropy with the reward from the environment. As shown in Fig. 5, Bilevel-LLM with entropy objective outperforms *Bilevel-LLM (Env)* and exhibits lower entropy of the aciton policy. **Can the Bilevel-LLM framework accommodate multimodal state representation?** We design a baseline *Bilevel-LLM-Symbolic*, where the action policy is replaced by that of Vanilla PPO, but taking both the embedding of the CoT output and symbolic environment observations as the input. As shown in Figure 3(b), Bilevel-LLM outperforms GFlan and *Bilevel-LLM-Symbolic* outperforms Vanilla-PPO, which indicates that the utilization of prompt questions and CoT reasoning is helpful to improve the capability of action policies with both textual and symbolic state representation.

## 5 RELATED WORK

**Reasoning with LLMs.** Previous studies have confirmed that stage-by-stage reasoning significantly enhances the capability of LLMs to solve complex tasks such as mathematical and logistic reasoning problems. Chain-of-Thought (CoT)(Wei et al., 2022) prompts containing a series of intermediate reasoning steps which is shown to improve the inference ability of

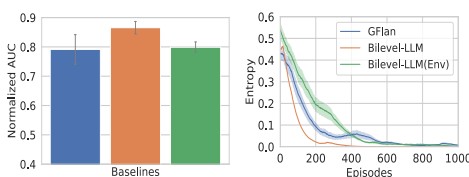

Figure 5: Ablation of the entropy objective on Chainworld (Partial). *Left:* Normalized AUC reward. *Right:* Entropy of the action policy.

LLMs. Self-consistency (Wang et al., 2022) marginalizes over several independent CoT reasoning paths and then selects the most consistent answer. PAL (Gao et al., 2023) integrates executable programs into the CoT reasoning, addressing computation-related problems. Besides using the prior world knowledge contained in LLMs, ReAct (Yao et al., 2022), Tree-of-Thought (ToT) (Yao et al., 2023) and RAP (Hao et al., 2023) make use of from external environments or internal LLMs to produce reasoning traces. ToT (Yao et al., 2023) and RAP (Hao et al., 2023) explore extensively compared to CoT. Both engage in multiple reasoning paths and construct a reasoning tree to determine the next crucial reasoning action through self-evaluation. In this work, LLMs are applied to address natural multi-step decision-making problems, such as the game of Overcooked, where rational reasoning is essential for each action.

**LLMs for RL.** Due to the impressive reasoning capabilities of humans and the wealth of knowledge preceding LLMs, a series of studies have attempted to incorporate LLMs into planning algorithms to address decision-making tasks. ICPI (Brooks et al., 2022) solves a number of simple interactive RL tasks (such as Maze) without the need for expert demonstrations or gradient computations, which is achieved by using LLMs as the world model and the rollout policy with historical interactions as in context examples. Chen et al. (2023) leverage historical trajectories to prompt LLM to generate the next step actions on the textWorld game. GFlan(Carta et al., 2023) aims to ground the LLM Flan-T5 (Rae et al., 2021) on solving a textual interactive task named BabyAI-Text. In this approach, Flan-T5 serves as the action policy and is fine-tuned via online PPO (Schulman et al., 2017). LFG (Shah et al., 2023), utilise an LLM with a polling strategy to recommend and subsequently rank subgoals. In our work, we integrate complex CoT reasoning with LLMs into RL to enhance interpretability and the value of each action while eliminating the need for meticulous engineering to interpret LLM outputs.

**Entropy in RL.** Entropy has been used extensively in RL as a tool for regularisation (Mnih et al., 2016; Albrecht et al., 2023). The policy in actor-critic methods is often trained with an additional term that aims to maximise the entropy of the learned actions, with the goal of exploring the environment without having a policy collapse early to suboptimal actions (e.g. Mnih et al., 2016). A more formal use of entropy is explored in maximum entropy reinforcement learning (Haarnoja et al., 2018; Eysenbach & Levine, 2021), where the optimisation objective aims to learn the optimal policy that has the maximum entropy. In this work, we take a different approach, and look at finding prompts that minimise the entropy of the action policy. Intuitively, this would push the CoT process to provide reasoning that makes the policy sure about its action. Such minimization of the entropy has also been explored: Zhang et al. (2021) formulate a hierarchical approach to intrinsic options, where entropy is minimised to improve the option sub-trajectories, and Allahverdyan et al. (2018) consider entropy for decision making in the exploration-exploitation trade-off.

**Automated Prompt Engineering.** The quality of prompts plays a crucial role in determining the output quality of LLMs. Many works hand-craft quality prompts such as the Generative Agents (Park et al., 2023) and ProAgent (Zhang et al., 2023). Apart from completely using human-crafted prompts, there are other studies that adopt different degrees of automation when generating meaningful prompts. For example, APE (Zhou et al., 2022) and DLN (Sordoni et al., 2023) generate prompts from multiple examples and utilise LLM to rank the prompt candidates. PromptPG (Lu et al., 2022) trained an additional prompt selection network using the policy gradient technique, where the deep network generation probability distribution over a predefined set of prompt examples. We also aim to minimise human-labor on prompt engineering, we therefore adopt the PromptPG method where we preset a group of prompts and let the algorithm choose for itself depending on the environment state.

## 6 CONCLUSION

We introduce Bilevel-LLM, a bilevel framework that is capable of learning introspective questions (in the form of prompts), then performing complex reasoning for guiding actions executed by an action-policy. The bilevel nature of the framework enables the accommodation of separate objectives for the two learning components, namely the prompt-generation policy uses an action-policy entropy minimisation objective which enables it to induce unambiguous and useful prompts to be fed to the action-policy. Meanwhile, the action-policy learns how to perform actions in the environment while making use of the CoT thoughts which it learns to interpret. We showed that this leads to a powerful framework that outperforms leading baselines in complex benchmark environments. We believe our framework takes an important step towards generalist artificial intelligence that is capable of introspection and complex decision-making.

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
