Figure 6: An example of the step-by-step inference process of Bilevel-LLM on the Overcooked task.

## 7 ADDITIONAL DESCRIPTION ABOUT THE METHODOLOGY

The following proof is a detailed derivation of the policy gradient result in equation 2 following the policy gradient method:

$$
\begin{aligned}
\nabla_\phi J(y|\pi_\theta, \pi_\phi, \pi^{re}) &= \sum_{\tau \sim \pi_\theta, \pi_\phi, \pi^{re}} \nabla_\phi \rho(\tau) R_o(\tau) \\
&= \sum_{\tau \sim \pi_\theta, \pi_\phi, \pi^{re}} \rho(\tau) \nabla_\phi \log \rho(\tau) R_o(\tau) \\
&= \sum_{\tau \sim \pi_\theta, \pi_\phi, \pi^{re}} \rho(\tau) \sum_{t \geq 0} \nabla_\phi \log \pi_\phi(p_t|o_t, \dots, o_{t-j\wedge 0}) R_o(\tau) \\
&= \mathbb{E}_{\tau \sim \pi_\rho, \pi_\theta, \pi^{re}} \sum_{t \geq 0} \nabla_\phi \log \pi_\phi(p_t|o_t, \dots, o_{t-j\wedge 0}) R_o(\tau) \\
&\approx \frac{1}{N} \sum_{t \geq 0} \nabla_\phi \log \pi_\phi(p_t|o_t, \dots, o_{t-j\wedge 0}) R_o(\tau) \\
&\approx \frac{1}{N} \sum_{t \geq 0} \nabla_\phi \log \pi_\phi(p_t|o_t, \dots, o_{t-j\wedge 0}) \hat{R}_t^o(\tau),
\end{aligned}
\tag{4}
$$

where $\hat{R}_t^o(\tau) := -\sum_{i \geq t} \left[ \gamma^{i-t} \mathcal{H}_\theta^\pi(y_i) | y_i = (o_i, v_{i+}), v_{i+} \sim \pi^{re}(i_t) \right]$ denotes the return-to-go from step $t$ to the end for the outer loop.

## 8 HYPERPARAMETER

As shown in Table 1,2,3, we report the important hyperparameters for learnable baselines Bilevel-LLM, GFlan and Vanilla PPO on all environments. Baselines GFlan and Vanilla PPO are trained by the PPO algorithm. For Bilevel-LLM, the prompt generation and action policies are alternately trained with the other policy frozen. The parameter update frequency indicates the number of collected episodes for each update. We do a grid search to select hyperparameters. We tune the learning rate in the range of [1e-3, 1e-4,1e-5], the batch size in the range of [16, 32, 64], and the update frequency in the range of [8,12,16,20]. We fix the training epochs for the action policy as 4, and epochs for the prompt generation policy as 8.

## 9 DETAILED DESCRIPTION OF EXPERIMENTS

### 9.1 ENVIRONMENT

**ChainWorld.** The ChainWorld game contains a linear sequence of states and the available actions for the agent are go left or go right. The agent gains a reward 100 at a random end of the chain and $-5$ at the other end, with $-1$ penalty for each move. At each episode, the award randomly appears on the left or right end, and the initial position of the agent is randomized, except for the ends. There

Table 1: The hyperparameters on ChainWorld

| Action policy | | | | |
|---|---|---|---|---|
| Baselines | Learning Rate | Epochs | Batch Size | Update Frequency |
| Bilevel-LLM | 1e-4 | 4 | 16 | 16 |
| GFlan | 1e-4 | 4 | 16 | 16 |
| Vanilla PPO | 1e-4 | 4 | 16 | 16 |
| Prompt generation policy | | | | |
| Baselines | Learning Rate | Epochs | Batch Size | Update Frequency |
| Bilevel-LLM | 1e-4 | 8 | 32 | 16 |
| GFlan | / | / | / | / |
| Vanilla PPO | / | / | / | / |

Table 2: The hyperparameters on FourRoom

| Action policy | | | | |
|---|---|---|---|---|
| Baselines | Learning Rate | Epochs | Batch Size | Update Frequency |
| Bilevel-LLM | 1e-5 | 4 | 32 | 16 |
| GFlan | 1e-5 | 4 | 32 | 16 |
| Vanilla PPO | 1e-5 | 4 | 32 | 16 |
| Prompt generation policy | | | | |
| Baselines | Learning Rate | Epochs | Batch Size | Update Frequency |
| Bilevel-LLM | 1e-4 | 8 | 32 | 16 |
| GFlan | / | / | / | / |
| Vanilla PPO | / | / | / | / |

Table 3: The hyperparameters on Overcooked

| Action policy | | | | |
|---|---|---|---|---|
| Baselines | Learning Rate | Epochs | Batch Size | Update Frequency |
| Bilevel-LLM | 1e-4 | 4 | 16 | 12 |
| GFlan | 1e-4 | 4 | 16 | 12 |
| Vanilla PPO | 1e-3 | 4 | 16 | 12 |
| Prompt generation policy | | | | |
| Baselines | Learning Rate | Epochs | Batch Size | Update Frequency |
| Bilevel-LLM | 1e-3 | 8 | 64 | 12 |
| GFlan | / | / | / | / |
| Vanilla PPO | / | / | / | / |

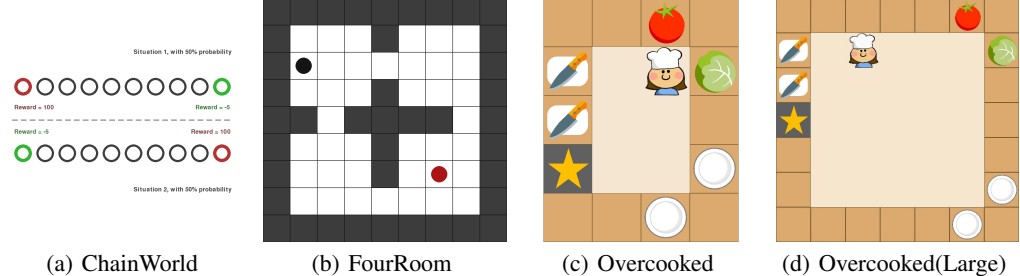

| (a) ChainWorld | (b) FourRoom | (c) Overcooked | (d) Overcooked(Large) |

Figure 7: Environment Diagrams. (a) Plots the ChainWorld Environment which has a 50% probability of the goal in the left end of the chain and 50% probability on the right. (b) The FourRoom Environment, which has four rooms and the position of the agent and the goal is randomly initialized. (c) The Overcooked Environment with a map size of $5 \times 4$. The agent needs to utilise the tools and material to make a salad and deliver it. (d) A large Overcooked layout with a map size of $7 \times 7$.

are two situations corresponding to different sides with high rewards. We consider two settings: *ChainWorld(Full)*, where full observation of the situation and position information are provided, and *ChainWorld(Partial)*, where only partial observation of the agent's position is available. In the case of ChainWorld(Partial), since the position with a reward of 100 is randomized, the agent must learn to make decisions based on historical trajectory information. The textual observations are as the following structure:

- *ChainWorld(Full)*: "You are at $<Position>$, 100 reward at $<Position>$."
- *ChainWorld(Partial)*:" You are at $<Position>$."

**FourRoom.** The FourRoom game is a navigation environment consisting of four rooms interconnected by four hallways in a circular arrangement. In this environment, the agent's initial position and the goal position are randomly set within these four rooms at the beginning of each game. This environment provides a textual description instead of symbolic observations. The brief textual description consists of a list of template descriptions as following structure:

- "You are in $<Room>$, goal is in $<Room>$."
- "The left-handed hallway's position is $<Position>$."
- "The right-handed hallway's position is $<Position>$."
- "Your position is $<Position>$. The goal's position is $<Position>$."

The $<Room>$ resembles one of the rooms, and the $<Position>$ is given as a list containing the x-coordinate and the y-coordinate. At each time step, the agent receives textual observations and decides whether to move in one of the following directions: up, down, left, or right within the grid. When the agent chooses to move, it incurs a penalty of $-2$ if the movement leads to an invalid location, such as attempting to traverse a wall or moving out of bounds. The agent is rewarded with 50 when it successfully reaches the goal, and receives a reward of $-0.4$ in other cases.

**Overcooked.** The Overcooked environment is a grid game where agent can move in four directions and interact with items in the map by moving towards item's direction when standing next it it. The goal for each agent is to make the correct meal and deliver to the star location. In our experiments, we used a text-based variant where the state information such as the name and location of each item, the location of the agent itself and the item the agent currently holds, are included in text:

- Currently in the kitchen there are the following items and their location:
    - Name: $<Item\ Name>$, Location: $<Item\ Location>$;
    - Name: $<Item\ Name>$, Location: $<Item\ Location>$;
    - ......
- textit$<$Agent Name$>$ is at location $<Agent\ Location>$ and currently holds $<Item\ Name>$

We use an incremental reward shaping scheme where we give small reward +1 to agent for fetching a correct material, medium reward +5 for successfully chopping the material, large reward +10 for

making a dish and putting it on a plat, finally a reward +100 for delivering the dish. The state space of Overcooked(Large) reaches $9.8 \times 10^{21}$ and state space of common Overcooked layouts with the map size $5 \times 4$ reaches $8.3 \times 10^{12}$.Take the Overcooked(Large) as an example. As shown in Figure 7, there are 21 positions on the kitchen counter where plates, tomatoes, lettuce, chopped tomatoes, chopped lettuce, a salad or nothing can be placed. Each cutboard can be in one of 5 states, which includes being empty, having one tomato, one lettuce, chopped tomato, chopped lettuce. The agent can stand at one of 25 grids and face one of four directions. Each player can hold either a plate, a tomato, a lettuce, chopped tomatoes, chopped lettuce, a completed salad or nothing in their hands. Therefore, the total number of possible states for this layout is: $7^{21}5^2C_{25}^1C_4^1C_7^1 \approx 9.8 \times 10^{21}$.

## 9.2 AUTOMATICALLY GENERATED PROMPT CANDIDATES FROM GPT3.5

**ChainWorld**    In the setting of fully observed ChainWorld(Full), we also do another experiments with the prompt candidates are automatically generated by the GPT3.5 relying on task's state and situation description.

- Task Description: In the ChainWorld game, an agent is situated on a line with 10 states, numbered from 0 to 9. The agent can take two actions: go left or go right. The episode ended when agent reached to the position with a reward score 100. Action: go left (-1), go right (1)

- State and situation: Current position: X(1 to 8) Position 0: reward -5 Position 9: reward 100 Reward -1 for each move. Please provide two prompt questions and thoughts about how to maximize reward.

- Prompts generated by GPT3.5:
  Prompt 1: How to maximize reward starting from position X to reach position 9 with a reward of 100?
  Prompt 2: What strategy should the agent use at position X to avoid position 0 with a reward of -5 while maximizing its overall reward?

- Thoughts generated by GPT3.5:
  Thought 1: To maximize the reward, consider taking the optimal sequence of go right actions.
  Thought 2: At position X, avoid go left toward -5, balance with go right to reach 100.

**Overcooked**

- Task Description: I would like you to help me work with an AI agent called "agent1" in a kitchen environment similar to the video game Overcooked. Inside the kitchen there are the following items: ["tomato", "lettuce", "plate0", "plate1","cutboard0", "cutboard1"]. There are also the following functions that you can use to make agent1 take actions:

  agent1.fetch(item: str) - go to the item's location and pick it up (item will be in agent1's hand), only the cutboard cannot be picked up or fetched
  agent1.put_onto(item: str) - put the object agent1 has in hand onto the item
  agent1.slice_on(item: str) - slice food (item has to be "cutboard0")
  agent1.deliver(None) - deliver the cooked food.
  Note that the agent can only hold one item at a time. Please provide three prompt questions and thoughts about how to maximize reward.

- Prompts generated by GPT3.5:
  Prompt 1: Ingredient Preparation How can agent1 efficiently fetch the "tomato" and "lettuce" from their initial locations to start the salad preparation process?
  Prompt 2: Assembly Process Once agent1 has the "tomato" and "lettuce" in hand, how should it proceed with assembling the salad on the plates ("plate0" and "plate1")?
  Prompt 3: Delivery Strategy What strategy should agent1 follow to deliver the prepared salads efficiently?

- Thoughts generated by GPT3.5:
  Thought 1: Consider the order in which the ingredients need to be fetched and whether any

preprocessing steps, such as slicing on the cutboard, are required.
Thought 2: Explore a sequence of actions involving fetching, putting onto the plates, and any other necessary steps to assemble a complete salad on each plate.
Thought 3: Consider factors such as the proximity of the plates to the delivery location and the ability of agent1 to carry only one item at a time. Think about optimal routes and whether any coordination is needed for efficient deliveries.

## 9.3 PROMPTS AND CoT EXAMPLES

### 9.3.1 CHAINWORLD

**Task Description:** There is a chain world game. A chain of states in it, the agent can go left or go right. An agent can gain the reward -5 in the one side of the chain and gain reward 100 in the otherside of the chain. How to prompt the agent to move to gain high reward please give 3 simple prompts when left side with 100 reward or right side or unknown side with 100 reward respectively

**Prompt Candidates for ChainWorld (Partial):**

1. Head left to discover a treasure trove of 100 points.

2. Go right to seize the opportunity for a generous 100-point prize.

3. Embrace the unknown, as 100 points could await in any direction.

### 9.3.2 FOURROOMS

**Task Description:** In a four room game, there are four rooms(0,1,2,3) connected by four hallways. To move through different rooms, agent can only go through the hallways. The agent can choose a left-handed manner or a right-handed manner to move through different rooms. The agent's initial position can be either in a room or in a hallway, and the goal position can be in any room. The objective for the agent is to reach the goal.

**Prompt Candidates:**

- You are in the same room as the goal, try to reach the goal.

- Goal is not in current Room; You should explore left-handed rooms to reach the goal.

- Goal is not in current Room; You should explore right-handed rooms to reach the goal.

- You are in the hallway between two Rooms. Go left-handed room, then explore.

- You are in the hallway between two Rooms. Go right-handed room, then explore.

**CoT Examples:**

1. Order: Goal is not in current Room; You should explore left-handed rooms to reach the goal.

   Observations: You in Room1. Goal in Room3. Entrance to Left-handed hallway: [6, 4]. Entrance to Right-handed hallway: [4, 6]. Goal position: [1, 1].

   CoT: "You in Room1, goal in Room3. Goal is not in current room. To move through different rooms, you can only go through hallways. Enter the left-handed hallway. How do you choose a step?"

2. Order: You are in the hallway between two Rooms. Go right-handed room, then explore.

   Observations: You in hallway. Goal in Room3. Entrance to Left-handed hallway: [6, 4]. Entrance to Right-handed hallway: [4, 6]. Goal position: [1, 1].

   CoT: "You in hallway. Goal is not in current hallway. Go to the right-handed room entrance. How do you choose a step?"

### 9.3.3 OVERCOOKED (SALAD)

**Task Description:** I would like you to help me work with an AI agent called "agent1" in a kitchen environment similar to the video game Overcooked. Inside the kitchen there are the following items: ["tomato", "lettuce", "plate0", "plate1","cutboard0", "cutboard1"]. There are also the following functions that you can use to make agent1 take actions:

agent1.fetch(item: str) - go to the item's location and pick it up (item will be in agent1's hand), only the cutboard cannot be picked up or fetched
agent1.put_onto(item: str) - put the object agent1 has in hand onto the item
agent1.slice_on(item: str) - slice food (item has to be "cutboard0")
agent1.deliver(None) - deliver the cooked food.
Note that the agent can only hold one item at a time.

**Prompt Candidates:**

1. Question: How would you instruct "agent-1" to start making a sliced lettuce? You do not need to fetch the cutboard nor delivering the food. Think about what tasks need to be accomplished step by step. task_queue = []

2. Question: How would you instruct "agent-1" to start making a sliced tomato? You do not need to fetch the cutboard nor delivering the food. Think about what tasks need to be accomplished step by step. task_queue = []

3. Question: With the sliced lettuce and the sliced tomato on the map and an empty hand, how would you instruct "agent-1" to make and deliver a lettuce-tomato salad? Think about what tasks need to be accomplished step by step. task_queue = []

**CoT Examples:**

1. task_queue = []
   # Step 1: Fetch a lettuce
   task_queue.append((agent1.fetch, "lettuce"))
   # Step 2: Put the lettuce onto the cutboard (assuming cutboard0 is available)
   task_queue.append((agent1.put_onto, "cutboard0"))
   # Step 3: Slice the lettuce on the cutboard
   task_queue.append((agent1.slice_on, "cutboard0"))

2. task_queue = []
   # Step 1: Fetch a tomato
   task_queue.append((agent1.fetch, "tomato"))
   # Step 2: Put the tomato onto the cutboard (assuming cutboard0 is available)
   task_queue.append((agent1.put_onto, "cutboard0"))
   # Step 3: Slice the tomato on the cutboard
   task_queue.append((agent1.slice_on, "cutboard0"))

3. task_queue = []
   # Step 1: Fetch a plate (choose either plate0 or plate1)
   task_queue.append((agent1.fetch, "plate0"))
   # Step 2: Put the sliced lettuce onto the plate
   task_queue.append((agent1.put_onto, "plate0"))
   # Step 3: Fetch the sliced tomato
   task_queue.append((agent1.fetch, "tomato"))
   # Step 4: Put the sliced tomato onto the plate
   task_queue.append((agent1.put_onto, "plate0"))
   # Step 5: Deliver the lettuce-tomato salad
   task_queue.append((agent1.deliver, None))

## 10 ADDITIONAL RESULTS

### 10.1 TRAINING PERFORMANCE

Figure 8 shows the training curves of the normalized rewards of the baselines (Bilevel-LLM, GFlan, Vanilla PPO). These results indicate that, on complex tasks except for ChainWorld, Bilevel-LLM significantly outperforms the other baselines and also exhibits lower variance.

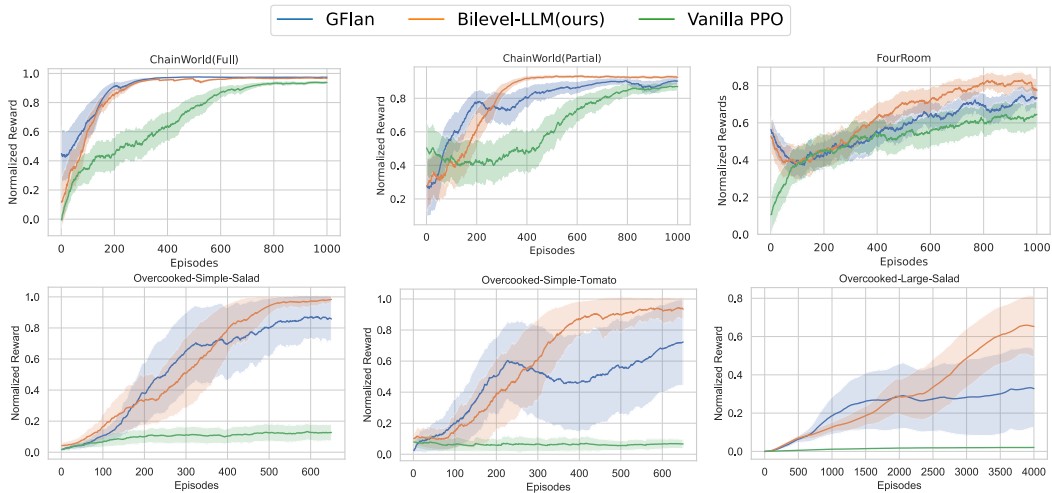

Figure 8: Training curves of baselines. We plot the average and standard error of normalized rewards over 5 seeds.

### 10.2 RESULTS ON CHAINWORLD

The performance of Bilevel-LLM, compared to other baselines in the ChainWorld environment is illustrated in Figure 9. Notably, as shown in Figure 9(c), Bilevel-LLM exhibits the most rapid reduction in action policy entropy. This observation suggests that Bilevel-LLM shows the potiential to effectively learn optimal prompts while guiding the action policy to focus on the optimal policies through the bi-level optimization. Consequently, as depicted in Figure 9(a) and Figure 9(b), Bilevel-LLM outperforms other baselines at the end of training as well as the AUC reward which represents the average performance throughout the training.

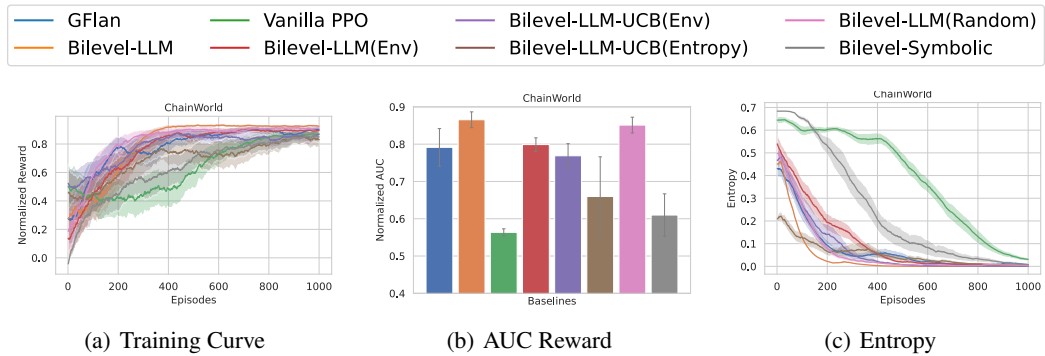

(a) Training Curve     (b) AUC Reward     (c) Entropy

Figure 9: Results on Chainworld(Partial). (a) Training curves of baselines. We plot the average and standard error of rewards over 5 seeds. (b) The average reward of baselines over all training episodes. (c) The entropy of action policy during training.

## 10.3 RESULTS ON FOURROOM

In the FourRoom environment, methods utilizing the LLM performs an increase of entropy in the first 200 episodes and turns to decay after 400 episodes. Examining Figure 10(a) and Figure 10(b) concurrently, an interesting trend emerges: the reduction in entropy often coincides with a decrease in reward. Conversely, an increase in entropy tends to correspond with a decline in performance, and the results supports the effectiveness of Bilevel-LLM.

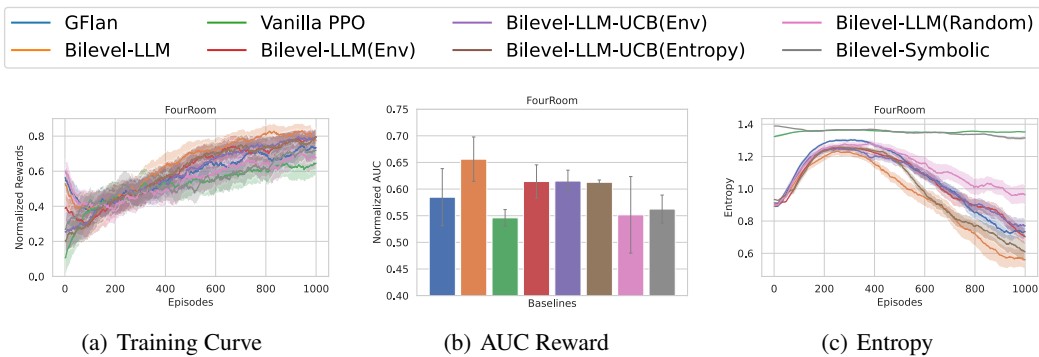

(a) Training Curve      (b) AUC Reward      (c) Entropy

Figure 10: Results on FourRoom. (a) Training curves of baselines. We plot the average and standard error of rewards over 5 seeds. (b) The average reward of baselines over all training episodes. (c) The entropy of action policy during training.

## 10.4 RESULTS ON OVERCOOKED

In the Overcooked environment, as shown in Figure 11, Bilevel-LLM exhibits the lowest action policy entropy demonstrates superior performance at the end of the training period. Similar to the previous environments, Bilevel-LLM shows its potential in mastering this game.

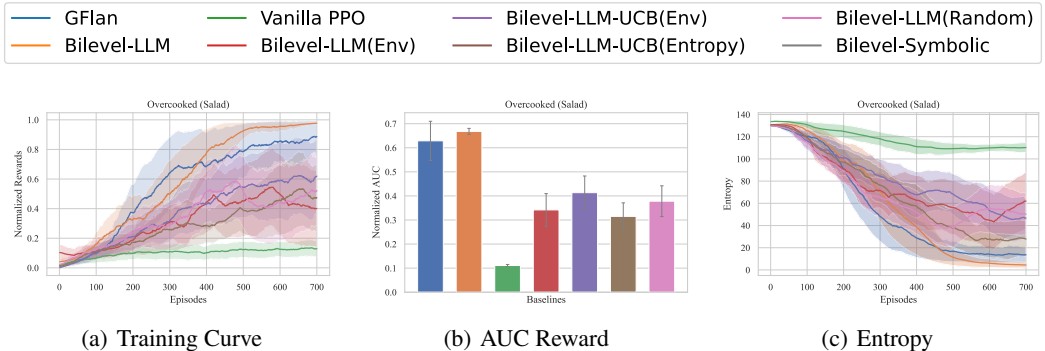

(a) Training Curve      (b) AUC Reward      (c) Entropy

Figure 11: Results on Overcooked (Salad). (a) Training curves of baselines. We plot the average and standard error of rewards over 4 seeds. (b) The average reward of baselines over all training episodes. (c) The entropy of action policy during training.