# OpenReview forum: "Ask more, know better: Reinforce-Learned Prompt Questions for Decision Making with  Large Language Models"
_ICLR.cc/2024/Conference — Submitted to ICLR 2024_

### Official Review · Reviewer_UBnq · 2023-10-26

**Soundness:** 1 poor
**Presentation:** 1 poor
**Contribution:** 1 poor
**Rating:** 3
**Confidence:** 5

**Summary:**

This paper proposes the leader-follower bilevel framework, for learning relevant prompts for task-relevant actions. To verify the effectiveness of the proposed method, experiments are conducted on the Overcooked and FourRoom environments.

**Strengths:**

1. The paper is well organized.
2. Figure 1 provides an intuitive illustration of the proposed method.

**Weaknesses:**

1. [Major] The presentation of this paper is vague and confusing. For example, this paper introduces many notations in the POMDP setting, but as far as the reviewer understands, the main usages of these notations only help the definition of policies $\pi_\phi,\pi^{re},\pi_\theta$. However, $\pi_\phi,\pi^{re},\pi_\theta$ themselves still remain unclear to the reviewer, even with the fancy math notations.  See other detailed examples in the “Question” section.
2. [Major] The experimental results are not convincing. After reading Section 4 (Experiments) and Appendix 8 (Detailed descriptions of experiments), the reviewer still could not understand how the experiments are conducted (e.g., what are the inputs and outputs of each environment, and what are the reward functions). The reviewer would like to see the inputs, outputs, and reward functions of each environment clearly introduced, not being vaguely described in the text. See detailed questions in the “Questions”.
3. [Minor] The paper seems to be overclaiming the contribution. In the first sentence of page 2, the author claims that “we take the first step towards a fully unified LLM framework that learns to perform complex tasks”. The reviewer does not believe the current paper is the “first” (see e.g., [1]), nor the Overvooked or FourRoom are complex enough (compared to [MineDojo](https://minedojo.org/) studied in [1]).

[1] Wang, Guanzhi, et al. "Voyager: An open-ended embodied agent with large language models." arXiv preprint arXiv:2305.16291 (2023).

**Questions:**

> The presentation of this paper is vague and confusing.

1. What are the exact inputs and outputs (e.g., are they symbolic vectors of the tasks texts) of $\pi_\phi,\pi^{re},\pi_\theta$? The reviewer has carefully read through section 2, and the reviewer guesses that (1) $\pi_\phi$ is a text-to-text mapping; (2) $\pi^{te}$ is a state-to-text mapping, but what exactly is the state space, is it text or images, or vectors? (3) $\pi_\phi$ is a mapping from (observation, text) to action, but what are the observation and action spaces (text, symbolic vectors, or others)?
2. In the last paragraph **CoT reasoning with Prompts**, the authors mentioned “$\pi^{re}$ is severed by an LLM such as GPT3.5”, what exactly is $\pi^{re}$? If it is GPT3.5, which version of GPT3.5 (`turbo`, `turbo-16k`, or others)? If not GPT3.5, what exactly is it?
3. In paragraph **Action policy training via PPO with LLM** of page 6, the authors mentioned “we use the pre-trained LLM, FLAN-T5 small… as the action policy”. Does this suggest that $\pi_\theta$ itself is also an LM? If yes, could the author clarify how the state space serves as text inputs to $\pi_\theta$ and how the text outputs of $\pi_\theta$ are post-processed into actions?

> The experimental results are not convincing.

1. What is the reward function in the `overcooked` environment? In Section 8.1 (Environment), the authors have introduced the reward functions for ChainWorld and FourRoom. But for Overcooked, the author mentioned in the last sentence of paragraph “Overcooked” that: “we use an incremental reward shaping scheme where we give a small reward to agent for…” What is the incremental reward shaping scheme, and how is the reward function actually defined in this case?
2. The experimental results presented in Figures 2 and 3 only slightly surpass the previous SOTA GFlan by a small margin, given the fact that only 5 random seeds are selected, it is hard for the reviewer to believe that the proposed method is actually better than GFlan. Note that in the original paper (Figure 3 of GFlan [1]), GFlan actually improves the prior baselines by a huge margin.


[1] Carta, Thomas, et al. "Grounding large language models in interactive environments with online reinforcement learning." arXiv preprint arXiv:2302.02662 (2023).

---

> ### Author Response · Authors · 2023-11-19
> **Response to Reviewer UBnq**
>
> We thank the reviewer for their time and effort in reviewing our paper. We have provided detailed explanations and clarifications which we believe resolve all your concerns and made several updates that have improved the paper based on your recommendations. With this, we hope you can increase your score to reflect these updates.
>
> > Q1: This paper claims "first step towards a fully unified LLM framework that learns to perform complex tasks"
>
> Our work provides a unified and **learnable** LLM framework for decision-making tasks, where **prompt generation and action policies are trainable**. We avoid the expensive prompt crafting in two ways. 1. We **train an action policy to automatically interpret the CoT reasoning and avoid human-crafted grounding functions** for mapping CoT reasoning to executable actions. 2. Our method successfully **learns to select from a global set of candidate prompts, the best prompt for each state**. The prompt candidate sets can generated in an automated fashion removing human hand-crafting. For more details, please refer to the discussion about the prompt generation policy on page 4 and the auto-generated prompts experiment illustrated in Fig. 3(c) and Fig. 4 on page 8. The Voyager environment provides an embodied lifelong learning agent by incorporating GPT-4 and a skill library, which does **not involve model parameter fine-tuning** but hand-crafted prompt engineering and retrieval techniques.
>
> In addition, we would like to state that Overcooked is considered to be a complex task due to its large state space and decision-making complexity and for this reason is used as a benchmark environment. We conducted **an additional experiment on a large Overcooked layout** with a map size of  $7\times 7$ and a recipe of tomato-lettuce salad, and **its state space reaches $9.8\times 10^{21}$**. For the reviewer's convenience, we provide the mean and standard error of normalised rewards in the table below (shows the results of Fig. 2 on page 7). The Vanilla PPO, GPT3.5, GPT3.5(CoT Prompt) failed to solve this task because of the large state space and the long-term and complex decision process.
>
> |                         | State Space          | Bilevel-LLM | GFlan   | Vanilla PPO | GPT3.5     | GPT3.5(CoT Prompt) |
> | ----------------------- | -------------------- | ----------- | ---------- | ----------- | ---------- | ------------------ |
> | Overcooked(Large,Salad) | $9.8\times10^{21}$ | 0.47(0.03)  | 0.31(0.08) | 0.03(0.01)  | 0.00(0.00) | 0.01(0.00)    |
>
> > Q2: The exact inputs and outputs of the prompt generation $\pi_\phi$, CoT reasoning $\pi^\text{re}$ and action $\pi_\theta$ policies
>
> The paper includes a description of each of these policies, please see the problem formulation section on page 3. We also provide an illustration of the detailed decision-making process of our method on Overcooked, which presents the involved inputs and outputs for each policy. Please see Fig. 6 of the Appendix on page 12.
>
> To explain here, the $\pi_\phi$ is the prompt generation policy, which takes symbolic vectors of historical observations as input and produces a distribution of prompt candidates as output. Then, a textual prompt question is sampled from this distribution. The details of the input, output, and training process of the prompt generation policy can be found in the methodology section on Page 5.
>
> The $\pi^{re}$ is the CoT reasoning policy. Its input is a textual prompt question and a textual situation description, and its output is a textual thought.
>
> The action policy $\pi_\theta$ can be either an LLM or a traditional MLP. For the LLM actor, please see the action policy training paragraph of the methodology section on page 6. For the MLP actor, please see the "Can the Bilevel-LLM framework accommodate multimodal state representation?" paragraph of the ablation study section on page 8. The input and output of the action policy depend on its type. If the action policy is an LLM, it takes the textual thoughts and observation description as input, and produces a distribution over textual actions as output. If the action policy is an MLP, it takes the sentence embedding of thoughts and symbolic observation representation as input, and produces a distribution over discrete actions as output. For more details about the input and output of the action policy, please see the answer to Q4.
>
> > Q3: The version of GPT3.5 (turbo, turbo-16k, or others)?
>
> The CoT reasoning policy is served by GPT3.5-turbo. We have added this detail in the revised version

---

> ### Author Response · Authors · 2023-11-19
> **Response to Reviewer UBnq**
>
> > Q4: If action policy is a pre-trained LLM?...If yes, could the author clarify how the state space serves as text inputs to and how the text outputs of are post-processed into actions
>
> Since the framework is plug-and-play, the action policy can be **either a pre-trained LLM to process text states or a traditional MLP to process symbolic states**. In this paper, we use the **LLM actor(action policy) for the main method (Bilevel-LLM)** and we conduct **an ablation study of the variant (Bilevel-LLM-Symbolic) that uses the MLP actor**. For the LLM actor, please see the action policy training paragraph of the methodology section on page 6. For the MLP actor, please see the "Can the Bilevel-LLM framework accommodate multimodal state representation?" paragraph of the ablation study section on page 8.
>
> When the action policy is an LLM, we can convert the symbolic state representation into text by using rule-based methods. Section 9.1 of the Appendix on page 14 shows some examples of textual state representations. The actions can also be described as text, such as "go left", "go right", and so on. We obtain the distribution over actions by applying a softmax operation on the probabilities of the action policy(an LLM) **outputting textual action descriptions**.
>
> When the action policy is an MLP, the input of the action policy consists of the sentence embedding of thoughts and the symbolic state representation.
>
> > Q5: What is the reward function in the overcooked environment?
>
> We have added the details of the reward shaping scheme for Overcooked, please refer to section 9.1 of the Appendix at the bottom of page 14.
>
> > Q6: Marginally surpasses GFlan
>
> The performance of our framework on ChainWorld is similar to GFlan **due to the environment simplicity of the problem setting**. Specifically, this environment has only **10 states and 2 actions** as illustrated in Figure 7(a) of the Appendix on page 15. We conducted an additional experiment on a larger Overcooked layout with a map size of  $7\times 7$ and a recipe of tomato-lettuce salad, whereas the common Overcooked layouts have a map size of $5\times 4$. The mean and standard error of AUC rewards (averaged over the whole training process) are shown in Fig.2 on page 7. The curves of normalised rewards during training processes are presented in Fig.8 of Appendix on page 18.
>
> For the reviewer's convenience, we provide the mean and standard error of normalised AUC rewards and the state space in the table below. **Our method outperformed GFlan and led to a smaller variance in both Overcooked and Fourroom tasks**. These results indicate that our framework improves the ability to solve complex decision-making tasks by **using the CoT reasoning that contains human priors and is triggered by appropriate prompt questions based on historical observations**. Please refer to section 9.1 of the Appendix on page 15 for the calculation details of the state space.
>
>
>
> |                         | State Space          | Bilevel-LLM | GFlan      |
> | -- | -- | ---- | ---------- |
> | FourRoom                | 1600                  | 0.66(0.04)  | 0.58(0.05) |
> | Overcooked(Salad)       | $8.3\times10^{12}$ | 0.65(0.01)  | 0.62(0.06) |
> | Overcooked(Tomato)      | $8.3\times10^{12}$ | 0.67(0.03)  | 0.52(0.10) |
> | Overcooked(Large,Salad) | $9.8\times10^{21}$ | 0.47(0.03)  | 0.31(0.08) |

---

> ### Author Response · Authors · 2023-11-21
> **Looking forward to further discussion**
>
> Dear reviewer UBnq,
>
> We sincerely appreciate the valuable comments on improving our paper. We hope that our responses sufficiently address your concerns. As the author-reviewer discussion period is coming to an end, please let us know if you have any further questions or concerns - we would be very happy to address them.

---

> ### Comment · Reviewer_UBnq · 2023-11-22
>
> Dear authors,
>
> Thank you very much for the updates. After carefully reading through the updated manuscript. I would like to hold my recommendation. Let me clarify:
>
> 1. I have no additional concerns about the experimental results.
>
> 2. But for the formulation part, the reviewer still believes there is something fundamentally unclear. See the following examples:
>
> - The reviewer did not find the author justifying why the formulations and notations adopt the POMDP setting. POMDP is much harder to study than MDP, which introduces additional notations and settings, making the formulation hard to interpret. For example, what is the state space $\mathcal{S}$ and what is the observation space $\mathcal{O}$? Are they text or are they symbolic vectors?
> - The reviewer did not find the updated manuscript self-contained, in terms of explaining what are the exact input-output of the policies $\pi_\theta,\pi^{re},\pi_\phi$.
>
> Note that the two examples above are not the only parts the reviewer found confusing. In fact, due to the overall presentation of the problem formulation setting, the reviewer cannot understand the contents in sections 2 & 3. Not even with the updated manuscript. Moreover, the reviewer did not find the updated manuscript to provide a better explanation of the formulation.
>
> The reviewer understands that this is an empirical paper, where the theoretical formulation may not be the major contribution, and the reviewer apologizes for the harsh nitpicking on the mathematical formulations.
>
> However, since the reviewer cannot verify the correctness/rigorousness of the formulation, the reviewer did not see an improvement in the clarity of the presentation. The reviewer cannot recommend acceptance.

---

> ### Author Response · Authors · 2023-11-22
> **Response to Reviewer UBnq**
>
> We thank the reviewer for their response and their time.
>
> In our update, we endeavored to carefully address each reviewer's questions and concerns.
>
> We would like to point the reviewer to parts of the paper that address the reviewer's concerns.
>
> For justification of the POMDP setting:
>
> Please see Section 9 in the updated supplementary material. There we explain that for example, in the Chainworld environment, we write: "We consider two settings: 1. ChainWorld(Full), where full observation of the situation and position information are provided, and 2. ChainWorld(Partial), where only partial observation of the agent’s position is available." There we also give specific details about these spaces.
>
> Therefore as some of the environments we consider are partially observable, a POMDP formulation is required.
>
> Since the exact form of the observation and state spaces varies between environments, we provided a general description of the POMDP setting in Section 2 when introducing the general problem setting. Specific details of these spaces are given in our experiment environments in Section 9.
>
>
> For justification of the text or symbolic representation of the state and observation space:
>
> We highlight that we consider the decision-making tasks to have a finite state space. Each state can be a symbolic vector that encodes the environmental features and can be translated into a text description that conveys the information in the vector. We have highlighted these explanations in Section 2 on page 3. Please refer to Section 9.1 of the supplemental material for examples of text descriptions. We have conducted experiments on both the symbolic and text representations. Please refer to the answer of Q4 for more details. Furthermore, for partially observable tasks, the observation only reveals partial information about the state, such as in the ChainWorld(Partial) task we explained earlier. For fully observable tasks, the observation contains all information of the state.
>
> For the specific inputs of each of the policies to provide further clarity for the reviewer, we updated the paper with an example of each input and output of each of the three policies. This is provided in Figure 6 on page 1 of our supplementary material.
>
> We hope this fully clarifies the reviewer's concerns and that our updates have now made the presentation of the paper as clear for reviewer UBnq as it now is for the other reviewers.

---

> ### Author Response · Authors · 2023-11-23
> **Response to Reviewer UBnq about the input-output of the policies**
>
> For the reviewer's convenience, we provide an example of the input-output of three polices on Overcooked here as shown in Figure 6 of our paper.
>
> * There is a predefined prompt candidate set $\mathcal{P}$ containing possible questions:
>
>     **prompt candidate set** $\mathcal{P}$: *1. How to slice lettuce? 2. How to slice tomato?3. How to deliver a lettuce-tomato salad?*
>
> *  At time step $t$, the environment provides:
>
>      **The observation $o_t$**: *Items X at Positions X. Agent is at location (3,2)*. *Agent currently holds nothing.*
>
>      **The situation $x_t$**: *Agent currently holds nothing.*
>
> * A prompt $p_t$ is selected from the prompt candidate set via the prompt generation policy $p_t\sim\pi_\phi(\cdot|o_t,\ldots,o_{t-j\wedge 0})$
>
>    **The selected prompt question** $p_t$: *How to slice lettuce?*
>
> * Get output of the CoT reasoning process  $\upsilon_{t}\sim\pi^{\rm re}(p_t,x_t)$
>
>   **The thought** $\upsilon_{t}$: *Step 1: Fetch a lettuce. Step 2: Put the lettuce onto the cutboard. Step 3: Slice the lettuce on the cutboard*.
>
> * An action $a_{t}\sim \pi_\theta(\cdot|o_t,\upsilon_{t})$ is taken.
>
>   **Action $a_{t}$:** Go left
>
> We hope this fully clarifies the reviewer's concerns and we respectfully hope you can consider your decision accordingly. If you have further concerns, please feel free to respond to us and we would like to discuss them with you.

---

### Official Review · Reviewer_g6Ja · 2023-10-29

**Soundness:** 3 good
**Presentation:** 4 excellent
**Contribution:** 3 good
**Rating:** 6
**Confidence:** 3

**Summary:**

This paper proposes the Bilevel-LLM approach for solving complex RL tasks. Bilevel-LLM (1) trains a promp-generating policy, (2) applies chain-of-thought reasoning to the prompt, and (3) trains an action policy conditioned on the final chain-of-thought output. The authors evaluate on several simple tasks (ChainWorld, FourRooms) and a more complex Overcooked task, where the authors demonstrate that their approach works better than several baselines.

**Strengths:**

- The paper is generally clearly written, with the algorithms and objective functions clearly written out.
- I believe the approach of using a policy to generate prompts for a CoT process that conditions an action policy is novel.
- The experiments and ablation studies convincingly show the value of the method. In particular, the ablation study over a random prompt generation baseline demonstrates the value of training the prompt generating policy, and the small performance increase over GFLan demonstrates the benefit of the more complicated bi-level approach.

**Weaknesses:**

- Some of the notation is complicated and I believe it could be simplified. In particular, there are many subscripts (t+, I, etc.) that I think could be removed that could make the presentaiton of the technical method a bit easier to read.

- The performance of the method is not that much higher than GFLan, which is surprising given that GFLan has no chain-of-thought reasoning. This implies that perhaps that chain-of-thought reasoning is not very effective for the tasks (given that it is a much more complex process), or that the tasks are too simple for the method to demonstrate it's benefits.

Minor:
The "t+" notation seems a bit unnecessary. I believe it can be removed and the equations would still make sense.

**Questions:**

- For reproducibility purposes, it would be great if the authors could report all learning rates & hyperparameters used in the experiments (also for baselines), as well as the hyperparameter sweeping strategy.
- What does the subscript "I" mean in the "\gamma_I" discount factor in Eq 1 & 4? I did not find this notation explained previously.

---

> ### Author Response · Authors · 2023-11-19
> **Response to Reviewer g6Ja**
>
> We thank the reviewer for their time and effort in reviewing our paper. We are glad that you appreciate the contributions of our method regarding the novel bilevel optimisation aspect, clear writing, and convincing experiments. We have provided detailed explanations and clarifications to resolve your concerns.
>
> > Q1: Notation
>
> The subscript t+ is used to discriminate the procedures from CoT reasoning at environment step t. We have made considerable effort to make the presentation as easy to digest as possible while ensuring that our expressions are mathematically sound. We have replaced the t+ by t for simplify as per your suggestion.
>
> > Q2: Performance as compared to to GFlan
>
> The performance of our framework on ChainWorld is similar to GFlan **due to the environment simplicity of the problem setting**. Specifically, this environment has only **10 states and 2 actions** as illustrated in Figure 7(a) of the Appendix on page 14. We conducted an additional experiment on a larger Overcooked layout with a map size of  $7\times 7$ and a recipe of tomato-lettuce salad, whereas the common Overcooked layouts have a map size of $5\times 4$. The mean and standard error of AUC rewards (averaged over the whole training process) are shown in Fig.2 on page 7. The curves of normalised rewards during training processes are presented in Fig.8 of Appendix on page 18.
>
> For the reviewer's convenience, we provide the mean and standard error of normalised AUC rewards and the state space in the table below. **Our method outperformed GFlan and led to a smaller variance in both Overcooked and Fourroom tasks**. These results indicate that our framework improves the ability to solve complex decision-making tasks by **using the CoT reasoning that contains human priors and is triggered by appropriate prompt questions based on historical observations**. Please refer to section 9.1 of the Appendix on page 15 for the calculation details of the state space.
>
>
>
> |                         | State Space          | Bilevel-LLM | GFlan      |
> | -- | -- | ---- | ---------- |
> | FourRoom                | 1600                  | 0.66(0.04)  | 0.58(0.05) |
> | Overcooked(Salad)       | $8.3\times10^{12}$ | 0.65(0.01)  | 0.62(0.06) |
> | Overcooked(Tomato)      | $8.3\times10^{12}$ | 0.67(0.03)  | 0.52(0.10) |
> | Overcooked(Large,Salad) | $9.8\times10^{21}$ | 0.47(0.03)  | 0.31(0.08) |
>
> > Q3: Learning rates and hyperparameters
>
> We report the important hyperparameters including learning rate, epoch, batch size, etc as per your suggestion. We do a grid search to select the optimal hyperparameters. For more details, please refer to section 8 of the Appendix on page 12.
>
> > Q4: The subscript "I" in the $\gamma_I$ discount factor (Eq 1 and 4)?
>
> $\gamma_I$ indicates the discount factor of the inner loop, i.e, the learning process of the action policy, we have explained the variable in the revision.

---

> ### Author Response · Authors · 2023-11-21
> **Looking forward to further discussion**
>
> Dear reviewer g6Ja,
>
> We sincerely appreciate the valuable comments on improving our paper. We hope that our responses sufficiently address your concerns. As the author-reviewer discussion period is coming to an end, please let us know if you have any further questions or concerns - we would be very happy to address them.

---

### Official Review · Reviewer_WqYj · 2023-10-30

**Soundness:** 2 fair
**Presentation:** 3 good
**Contribution:** 2 fair
**Rating:** 6
**Confidence:** 4

**Summary:**

This paper presents a bilevel framework that consists of the prompt-generation policy, thought reasoning policy, and action policy. In particular, at each timestep, the prompt-generation policy generates a prompt, which is then used by the thought reasoning policy to produce the CoT. The produced thought is used by the action policy to select an action in the environment. The proposed bilevel-LLM method is evaluated in ChainWorld, FourRoom, and Overcooked domains and shows better performance than baselines.

**Strengths:**

1. The bilevel optimization with the prompt-generation policy to minimize the actor policy's uncertainty is new & interesting.
2. The paper is well-written and addresses the important challenge of automated prompt engineering for solving decision-making tasks.

**Weaknesses:**

1. While the proposed bilevel-LLM generally achieves better performance than baselines, the performance gap is marginal compared to the GFlan baseline (the difference may not be statistically significant).
2. The novelty could be limited with respect to prior work: learning to select a prompt based on policy gradient is similar to PromptPG, and the use of Chain-of-thought prompting to act in the environment is similar to ReAct. As such, this paper could be viewed as combining these two directions.
3. While one of the paper's objectives is to avoid the expensive prompt crafting by humans, the framework would still need human-constructed questions. In Figure 3(c), the paper presents the Bilevel-LLM-Auto that does not rely on human-constructed prompts, but it is unclear whether this method applies to other domains, including Overcooked (only the ChainWorld(Full) performance is shown).

**Questions:**

1. I hope to ask the authors' responses to my concerns (please refer to the weaknesses section for details).
2. The prompt-generation policy aims to minimize the actor policy's entropy. However, I am unsure whether this is the correct objective to optimize for because 1) the actor policy may be certain but certain about incorrect actions (i.e., low uncertainty but convergence to sub-optimal actions), 2) for some domains, an optimal policy could be stochastic not deterministic, and 3) a positive entropy could help exploration.
3. The proposed bilevel optimization could be difficult because the actor policy is learning and thus keeps changing its behavior over time. Theoretically, this non-stationary actor's behavior makes the reward function (i.e., the entropy reward) non-stationary from the prompt-generation policy's perspective, which could render the Markov property invalid and induce unstable training of the prompt-generation policy. Would it be possible to ask for more discussion about this possible non-stationarity issue, which is one of the main challenges in multi-agent learning?
4. Because the domain is POMDP (Section 2), would the Vanilla PPO use RNN/LSTM/transformer architecture instead of MLP architecture (i.e., no memory)?

---

> ### Author Response · Authors · 2023-11-19
> **Response to Reviewer WqYj**
>
> We thank the reviewer for their time and effort reviewing our paper. We are glad that you appreciate the contributions of our method regarding the novel bilevel optimisation aspect and the challenge of automated prompt engineering for decision-making tasks. We have provided detailed explanations and clarifications to resolve your concerns. We hope you can increase your score to reflect our updates.
>
> > Q1: Performance compared to the GFlan baseline
>
> The performance of our framework on ChainWorld is similar to GFlan due to the simplicity of the problem setting. Specifically, this environment has only **10 states and 2 actions** as illustrated in Figure 7(a) of the Appendix on page 15. We conducted an additional experiment on a larger Overcooked layout with a map size of  $7\times 7$ and a recipe of tomato-lettuce salad, whereas the common Overcooked layouts have a map size of $5\times 4$. The mean and standard error of AUC rewards (averaged over the whole training process) are shown in Fig.2 on page 7. The curves of normalised rewards during training processes are presented in Fig.8 of Appendix on page 18.
>
> For the reviewer's convenience, we provide the mean and standard error of normalised AUC rewards and the state space in the table below. **Our method outperformed GFlan and led to a smaller variance in both Overcooked and Fourroom tasks**. These results indicate that our framework improves the ability to solve complex decision-making tasks by **using the CoT reasoning that contains human priors and is triggered by appropriate prompt questions based on historical observations**. Please refer to section 9.1 of the Appendix on page 14 for the calculation details of the state space.
>
>
>
> |                         | State Space          | Bilevel-LLM | GFlan      |
> | -- | -- | ---- | ---------- |
> | FourRoom                | 1600                  | 0.66(0.04)  | 0.58(0.05) |
> | Overcooked(Salad)       | $8.3\times10^{12}$ | 0.65(0.01)  | 0.62(0.06) |
> | Overcooked(Tomato)      | $8.3\times10^{12}$ | 0.67(0.03)  | 0.52(0.10) |
> | Overcooked(Large,Salad) | $9.8\times10^{21}$ | 0.47(0.03)  | 0.31(0.08) |
>
> > Q2: Our framework in relation to a combination of PromptPG and ReAct.
>
> The promptPG framework focuses on solving a supervised learning task. It uses the accuracy of predicting one label as a reward. A key novelty of our work is **the bilevel framework which allows training prompt-generation and an action policy concurrently** while allowing to prompt-generation training to take account of the behaviour of the action policy. Additionally, in order to generate greater certainty in the output of the CoT reasoning output, we use the (negative) entropy of the action policy as the reward to train the prompt-generation policy. This to our knowledge is a novel use of entropy minimisation in this setting. Different from React, which uses a **frozen LLM (PaLM-540B)** to generate domain-specific actions by **in-context examples**, the action policy in our framework is **trained to achieve optimal performance**, specifically, it is trained using RL on the environment reward. Another key point of distinction is that unlike PromptPG and React **our framework uses interactions with the environment using a trainable action policy**.
>
> > Q3: ...Avoidance of expensive manual prompt crafting and Bilevel-LLM-Auto in other domains
>
> Our work provides a unified training framework for decision-making tasks integrating prompt generation, CoT reasoning, and action policies. We avoid the expensive prompt crafting in two ways.
>
> 1. We train **an action policy to automatically interpret the CoT reasoning and avoid human-crafted grounding functions** for mapping CoT reasoning to executable action. A detailed description of the action policy appears on page 6 which we refer the reviewer to.
>
> 2. Our method successfully **learns to select from a global set of candidate prompts, the best prompt for each state**. The prompt candidate sets can generated in an automated fashion removing human hand-crafting. We refer the reviewer to the discussion about the prompt generation policy on page 4 and the auto-generated prompts experiment illustrated in Fig. 3(c) and Fig. 4 on page 8. We have verified the auto-generated prompt candidates from GPT-3.5 on ChainWorld and have updated the paper with further examples of this (auto-generated prompt candidates) in Overcooked.

---

> ### Author Response · Authors · 2023-11-19
> **Response to Reviewer WqYj**
>
> > Q4: The prompt-generation objective
>
> A key point to note is that the action policy is (independently) being updated towards actions that increase the return from the environment. This is in accordance with its own objective which is to maximise its environment return. This happens while the prompt generation policy observes the behaviour of the action policy during its training process. Under standard assumptions, the action policy converges to a local maximum of its objective which is to maximise the expected environment return.
>
> The (negative) entropy is the objective of the prompt generation policy so as to produce prompt questions that make the action policy more confident about its decisions given the input from the CoT output. As shown in Fig. 5 on page 8, we can see that our method (using negative entropy as prompt generation policy's objective) outperforms the variant Bilevel-LLM(env) that uses environment rewards as the reward for the prompt generation policy. Our also method achieves lower action policy entropy than Bilevel-LLM(env), which indicates that the entropy objective is helpful in reducing the uncertainty of the action policy.
>
> > Q5: Non-stationarity during training and relation to multi-agent learning.
>
> The reviewer makes an interesting point. We think this important question is worth further investigation in future (theoretical) analyses. In our numerous experiments, we observed convergence each time. We have two points of intuition for this. First, the prompt generation policy observes the behaviour of the action policy which is **analogous to centralised training in multi-agent reinforcement learning**. Second, **the two policies are alternatively trained** with one policy being trained while the other policy is frozen. This induces a two-timescale system [1]of training. Such systems are known to have convergence guarantees under certain conditions. Please refer to lines 12 to 18 of Algorithm 1 on page 5 for the alternative training scheme.
>
> [1]Borkar, V. S., \& Borkar, V. S. (2008). Multiple Timescales. Stochastic Approximation: A Dynamical Systems Viewpoint, 64-77.
>
> > Q6:  would the Vanilla PPO use RNN/LSTM/transformer architecture instead of MLP architecture (i.e., no memory) for the POMDP setting?
>
> For the environment ChainWorld(partial) which is a POMDP, we use the historical observations (with a length of 20) as input for the action policy. The React you recommended and DVRL[2] are also based on historical context to infer the next step action.
>
> [2]Igl, Maximilian, et al. "Deep variational reinforcement learning for POMDPs." International Conference on Machine Learning. PMLR, 2018.

---

> ### Author Response · Authors · 2023-11-21
> **Looking forward to further discussion**
>
> Dear reviewer WqYj,
>
> We sincerely appreciate the valuable comments on improving our paper. We hope that our responses sufficiently address your concerns. As the author-reviewer discussion period is coming to an end, please let us know if you have any further questions or concerns - we would be very happy to address them.

---

> ### Author Response · Authors · 2023-11-21
> **Providing additional experiment results on applying automatically generated prompt candidates by GPT3.5 for the Overcooked**
>
> We thank the reviewer for their comments and valuable suggestions. To fully address the reviewer's query we have added more experiment results on applying automatically generated prompt candidates by GPT3.5 in the Overcooked environment.
>
> Please see Fig. 4 and "Does Bilevel-LLM learn to automatically generate prompts" paragraph on page 8 for more details. For the reviewer's convenience, we provide the mean and standard error of normalised **testing rewards after baselines are trained on the same number of episodes** and the same episodes in the table below which indicate that our auto-prompt generation end-to-end framework successfully tackles the task with high performance.
>
>
>
> |                   | State Space          | Bilevel-LLM | GFlan      | Bilevel-LLM-Auto |
> | ----------------- | -------------------- | ----------- | ---------- | ---------------- |
> | Overcooked(Salad) | $8.3\times10^{12}$ | 0.99(0.01)  | 0.90(0.08) | 0.96(0.03)       |
>
> Bilevel-LLM-Auto (a variant of our method that applies automatically generated prompt candidates by GPT3.5) **achieves similar performance to Bilevel-LLM** (our main method with human-designed prompt candidates). Moreover, Bilevel-LLM and Bilevel-LLM-Auto both **surpass GFlan and exhibit lower variance. Specifically, after training for the same number of episodes, Bilevel-LLM attains a normalized reward of about 1.0, while GFlan only reaches around 0.9. This suggests that CoT thoughts induced by appropriately selected prompts are beneficial for solving the complex decision-making task and that **our plug-and-play framework** is able to learn to use the automatically generated valuable prompts from GPT3.5.
>
> We hope that our responses have solved your concerns and respectfully hope you can increase your score accordingly. As the review period is soon to come to a close we would appreciate it if you could confirm we have resovled your query or if you have any concerns (which we would be happy to address).

---

> > ### Comment · Reviewer_WqYj · 2023-11-22
> > **Response to Rebuttal**
> >
> > I appreciate the authors for the detailed response to my feedback and for generating additional results. Please find my response below:
> >
> > Q1: Performance compared to the GFlan baseline.
> > Thank you for the new results of Figures 2 and 8. Compared to the original paper, the new revision shows larger margins compared to GFlan.
> >
> > Q2: Our framework in relation to a combination of PromptPG and ReAct.
> > Thank you for the clarification. I agree with the bilevel framework point.
> >
> > Q3: ...Avoidance of expensive manual prompt crafting and Bilevel-LLM-Auto in other domains.
> > Thank you for the new result of Figure 4. Ideally, it would be great to add results for Overcooked Tomato and Large.
> >
> > Q4-6: Thank you for the clarifications.
> >
> > The new results (partially) address my concerns, so I have updated my score accordingly. If this paper is accepted, GFlan's performance in Figure 8 is a little noisy, so I hope to increase the number of seeds to 10 for improved paper quality.

---

> ### Author Response · Authors · 2023-11-23
> **Response to Reviewer WqYj**
>
> We thank the reviewer for their positive response and their time.
>
> Q3: add Bilevel-LLM-Auto results for Overcooked Tomato and Large.
>
> Thanks for your suggestion, we will add Bilevel-LLM-Auto ablation experiments for Overcooked Tomato and Large in the final version.
>
>
> Q: GFlan's performance in Figure 8 is a little noisy
>
> For all baselines, we trained 5 models with different random seeds for fair comparisons. Our method exhibited lower variance than GFlan, which may be attribute to the CoT reasoning that incorporates rich human prior and reduces the exploration needs, and the prompt generation policy that aims to reduce the action policy uncertainty. And we will train with more random seeds for more roubst results in the final version.

---

### Official Review · Reviewer_5KxL · 2023-11-01

**Soundness:** 3 good
**Presentation:** 2 fair
**Contribution:** 2 fair
**Rating:** 5
**Confidence:** 4

**Summary:**

This paper proposes a bi-level framework which learns to select a set of prompts to enable effective operation on downstream tasks. A set of prompts is generated using chatGPT 3.5, which are then selected from using a learned policy. Generated chain of thought skeleton are then input into a action policy which acts in the environment. Experiments show the efficacy of the approach.

**Strengths:**

- The proposed bi-level mechanism to select from a set of prompts and then act in the environment is novel to my understanding
- The method in the paper I generally clear and understandable

**Weaknesses:**

- The formatting of the paper is odd, for example the distance from the subsection heading from text is way too small in page 5
- The results in the paper would be more readable if there were more illustrations of the process (for example in the introduction)
- The LLM takes as input only text. In this case, for embodied domains with image observations, it seems like there is no way for the LLM to really know the current state, which means that the method essentially just trains a low-level image based policy given some set of thoughts (since the high-level policy is invariant to task completion).
- As a result, this framework doesn't really make sense for decision-making in my opinion -- it makes much more sense in the setting of reasoning and I would like to see evaluation in that setting.

**Questions:**

- Given the setting described above, why is that the approach actually improve performance over baselines?

---

> ### Author Response · Authors · 2023-11-19
> **Response to Reviewer 5KxL**
>
> We thank the reviewer for their time and effort in reviewing our paper, We are glad that you appreciate the main advantages of our method. We have provided detailed explanations and clarifications to resolve your concerns and made several updates that have improved the paper based on your recommendation. We respectfully hope you can reconsider your score to reflect our updates.
>
> > Q1: The paper formatting
>
> Thank you for pointing out the issue on formatting, we have adjusted the formatting of the paragraph title on page 5.
>
> > Q2: Request for more illustrations
>
> We agree with the reviewer's comment that illustrations are very useful. For this reason, we have illustrated the proposed Bilevel optimisation framework and an example of prompt candidates and CoT reasoning in an overcooked environment as shown in Figure 1 on page 3. We further add a more detailed decision inference process following our framework on Overcooked as per your suggestion, which presents the detailed inputs and outputs for each component of our framework. This is presented in Figure 6 of the Appendix on page 12 which we will include it in the main body in the final version.
>
>
> > Q3: Input observations and application in domains with image observations
>
> Our main goal is to verify the effectiveness of our framework in some well-known RL benchmarks. A large number of standard RL benchmarks of representative decision-making tasks, for example, StarCraft, Overcooked, and Card games have **symbolical observation representations that can be converted into text descriptions via rule-based methods**. In this work, we have verified the effectiveness of our framework on several decision-making tasks with symbolic observation representation, including ChainWorld, FourRoom, and Overcooked. We present examples of textual observation representations on these games in section 9.1 of the Appendix on page 14. In decision-making scenarios with image-form observations, such as autonomous driving and embodied robotics, a number of well-established techniques exist for converting image inputs to text representations, such as GPT-4 Vision and Minigpt-4[1]. Since these techniques can be easily integrated into our framework, our framework ought to be straightforwardly made to work in settings with image-based observations.
>
> [1]Zhu, Deyao, et al. Minigpt-4: Enhancing vision-language understanding with advanced large language models. arXiv preprint arXiv:2304.10592 (2023).
>
> > Q4: Decision-making versus reasoning tasks
>
> Our framework focuses on solving the decision-making tasks by incorporating prompt generation, CoT reasoning, and action policies. The prompt generation and CoT reasoning policies aim to generate high-level guidance by reasoning about the tasks. For example, LLMs infer the agent should walk to a tomato in the task Overcooked. The action policy $\pi_\theta$ is employed to perform executable actions, specifically, it should go left first to approach the target. However, **some reasoning tasks cannot be easily converted into Markov Decision Processes**, such as the well-known mathematics reasoning dataset GSM8K since they do not have obvious action spaces and transition dynamics.
>
> > Q5:Why our approach improves performance over baselines?
>
> We would like to first point out that a key contribution of our framework is to remove the necessity of manual engineering for interpreting CoT reasoning and grounding it into action space. With this, we can in fact surpass the performance of existing approaches. *1.* Compared with traditional RL baselines, our method incorporates the CoT reasoning induced by prompt questions to guide the action policy to perform more confidently. By using CoT reasoning, the action policy can **leverage high-level reasoning and knowledge captured with the LLM**, which leads to improved performance and fewer exploration requirements over RL baselines. An in-depth discussion on this can be found in our introduction on page 2.
> *2.* Additionally, reasoning baselines such as GPT3.5 (with CoT prompt) may not solve decision-making tasks well since **they require grounding and human-engineered grounded functions which are both prone to human error and fail to generalise**.
> (more details of this are discussed in the introduction on page 1). Our framework outperforms these baselines by employing reinforcement learning to autonomously and optimally ground the observation and CoT reasoning into executable actions. As shown in Figure 2 on page 7, our method outperforms RL baselines and reasoning baselines (i.e., GPT3.5 without CoT and GPT3.5 with CoT reasoning) on almost all environments as well as exhibiting a much smaller variance.

---

> ### Author Response · Authors · 2023-11-21
> **Looking forward to further discussion**
>
> Dear reviewer 5KxL,
>
> We sincerely appreciate the valuable comments on improving our paper. We hope that our responses sufficiently address your concerns. As the author-reviewer discussion period is coming to an end, please let us know if you have any further questions or concerns - we would be very happy to address them.

---

> ### Comment · Reviewer_5KxL · 2023-11-28
> **Rebuttal Response**
>
> I thank the authors for their response to the rebuttal -- I remain quite concerned about the applicability of the approach to settings with images, which make up a vast amount of the decision settings we consider. I would like to maintain my current rating for now and encourage the authors in future work to either alleviate this limitation or tone down the introduction and method section to explicitly acknowledge this limitation (which is rather major).

---

### Author Response · Authors · 2023-11-21
**Global Response**

Dear reviewers

Thank you once more for your efforts and valuable insights during the review process. We believe we have comprehensively responded to your inquiries and revised our manuscript to include your suggestions. In particular, we have made the following changes:

*1.* We conducted an additional experiment on a large overcooked layout (with a state space of $9.8 \times 10^{21}$). On this complex task, our method significantly outperformed other baselines. Please refer to Figure 2 on page 8 and Figure 8 in the Appendix on page 19 for more details.

*2.* We conducted an additional experiment on applying the auto-generated prompt candidates from GPT3.5 for the Overcooked task. Please see Figure 4 and detailed analysis on page 8.

*3.* We added a detailed illustration to better clarify our framework, where the inputs and outputs of each component are exemplified. Please refer to Figure 6 of the Appendix on page 13 for more details.

*4.* We have added more details about the environment setting in the Appendix, such as the hyperparameters, the reward shaping scheme, some examples of auto-generated prompt candidates, and the highlighted textual observation representations.

The changes to the main text are shown in blue for the reviewers' convenience.
We would appreciate it if each reviewer could respond to our answers and confirm that you are now satisfied with the updates or if there are any outstanding questions.

---

### Meta-Review · Area_Chair_Xpft · 2023-12-09

**Metareview:**

The paper introduces a bi-level framework for prompt selection in decision-making tasks and is evaluated in various environments.
The strength lies in 1) The paper introduces a novel bi-level framework for prompt selection in decision-making tasks.
2) The method is generally well-written and clear, with an attempt to address the challenge of automated prompt engineering.
3) The paper acknowledges the significance of learning prompt relevance for solving decision-making tasks.

The weakness lies in 1) Overclaims are identified, with a dispute regarding being the "first" in the field.
2) Concerns about the applicability of the approach to settings with images and the need for evaluation in reasoning settings.
3) Similarities to prior work in prompt selection are pointed out, suggesting potential limitations in novelty.

In summary, while there is acknowledgment of the novelty and potential of the proposed framework, some reviewers express significant concerns about clarity, presentation, experimental results, and potential overclaims. Addressing these issues is crucial for the paper's consideration. Therefore at current stage, the discussion leads to rejection of this paper.

**Justification For Why Not Higher Score:**

All the reviewers were not excited about this paper. The novelty seems incremental.

**Justification For Why Not Lower Score:**

N/A

---

### Decision · Program_Chairs · 2024-01-16

Reject